# Interesting Object, Curious Agent:
# Learning Task-Agnostic Exploration

**Simone Parisi**[1*]   **Victoria Dean**[2*]   **Deepak Pathak**[2]   **Abhinav Gupta**[1]
[1]Facebook AI Research   [2]Carnegie Mellon University

## Abstract

Common approaches for task-agnostic exploration learn tabula-rasa –the agent assumes isolated environments and no prior knowledge or experience. However, in the real world, agents learn in many environments and always come with prior experiences as they explore new ones. Exploration is a lifelong process. In this paper, we propose a paradigm change in the formulation and evaluation of task-agnostic exploration. In this setup, the agent first *learns to explore* across many environments without any extrinsic goal in a task-agnostic manner. Later on, the agent effectively transfers the learned *exploration policy* to better explore new environments when solving tasks. In this context, we evaluate several baseline exploration strategies and present a simple yet effective approach to learning task-agnostic exploration policies. Our key idea is that there are two components of exploration: (1) an agent-centric component encouraging exploration of unseen parts of the environment based on an agent's belief; (2) an environment-centric component encouraging exploration of inherently interesting objects. We show that our formulation is effective and provides the most consistent exploration across several training-testing environment pairs. We also introduce benchmarks and metrics for evaluating task-agnostic exploration strategies. The source code is available at https://github.com/sparisi/cbet/.

## 1 Introduction

Exploration is one of the key unsolved problems in building intelligent agents capable of behaving like humans. In reinforcement learning (RL), exploration is usually studied under two different settings. The first is task-driven exploration, where the reward is well-defined and the agent's goal is to explore in order to maximize long-term rewards. However, in real life, external rewards are either sparse or unknown altogether. In this setting, exploration is task-agnostic: given a new environment, the agent has to explore it in absence of any external reward. Common approaches to encourage task-agnostic exploration use intrinsically motivated rewards such as prediction curiosity [35, 47], empowerment [39], or visitation counts [4, 34]. But does this setup represent how humans explore?

We argue that the commonly-used task-agnostic exploration setup is unrealistic, both from practical and academic viewpoints. This setup assumes environments in isolation and agents exploring tabula-rasa, i.e., with no prior knowledge or experience. By contrast, we as humans do not learn from one environment in isolation and we do not throw away our past knowledge every time we encounter a new environment [14]. Exploration is rather a lifelong process: every time we encounter new environments, we use our prior knowledge and experience to develop new efficient exploration strategies. In this paper, we view the exploration problem from a continual learning lens. More specifically, in this setup, the learning agent interacts with one or many environments without any extrinsic goal. At this time, the agent *learns to explore* the environments. Later on, the agent effectively transfers the learned *exploration policy* to explore new environments, rather than exploring the new environment tabula-rasa.

---

*Equal contribution. Contacts: sparisi@fb.com and vdean@cmu.edu

35th Conference on Neural Information Processing Systems (NeurIPS 2021).

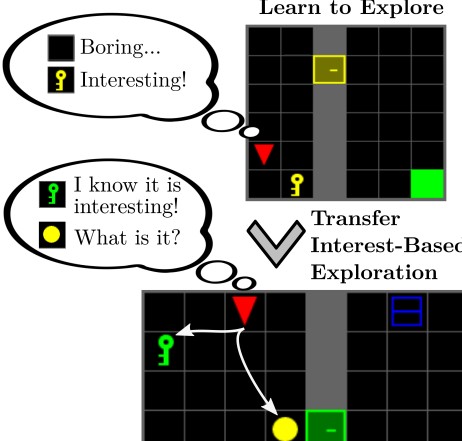

Figure 1: **Change-Based Exploration Transfer (C-BET)** trains task-agnostic exploration agents that transfer to new environments. Here the agent learns that keys are interesting, as they allow further interaction with the environment (opening doors). Later, when tasked with reaching a box behind a door, the agent starts by picking up the key.

A key question in learning how to explore is what to learn and how to transfer prior knowledge from one environment to another. Most existing task-agnostic exploration approaches, such as visitation counts, curiosity, or empowerment, define intrinsic rewards in an *agent-centric* manner: they encourage exploration of unseen parts of the environment based on the agent's own belief. In these approaches, exploration is driven by what the agent knows about the world. However, most do not make a distinction between what the agent believes it is interested in and states that would make any agent interested. For example, if the agent uses a visitation count model and has seen many objects of one kind in one environment, it would not explore the same type of objects again in a new environment. This seems to be in stark contrast to how humans explore. Consider a switch with a bell sign. Even though we might have pressed hundreds of doorbell switches (and even this instance), we are still attracted to press it. Some objects in the world just demand curiosity. We argue that apart from an 'agent-centric' component, there is an 'environment-centric' component to exploration, which can be learned from prior knowledge and experiences.

In this paper, we propose a paradigm change to move away from stand-alone isolated task-agnostic environment exploration to a more realistic multi-environment transfer-exploration setup[2]. We show how to learn exploration policies both from single- and multi-environment interaction, and how to transfer them to unseen environments. This transfer-exploration setup allows agents to use prior experiences for learning task-agnostic exploration. Notably, classic stand-alone task-agnostic approaches were designed for tabula-rasa exploration and hence only explore in an agent-centric manner. They fail to capture the inherent interestingness of some environment components. With this insight, we propose *Change-Based Exploration Transfer (C-BET)*, a simple yet effective approach learning joint agent-centric and environment-centric exploration. The key idea is for an agent to seek out both surprises (unseen areas) and high-impact (interesting) components of the environment. The experiments show that C-BET (a) learns more effectively when placed in a multi-environment setup, and (b) either outperforms or performs competitively with prior methods across several unseen testing environments. We hope this paper will inspire exploration research to focus more on learning from multiple environments and transferring experiences rather than tabula-rasa exploration.

## 2 Preliminaries and Related Work

We consider environments governed by Markov Decision Processes (MDPs). In MDPs, an agent observes the state of the environment $s$ and selects actions $a$ according to a policy $\pi(a|s)$. In turn, the environment changes, providing a new observation $s'$ and a reward $r$. Through environment interaction, the agent collects episodes, i.e., sequences of states, actions and rewards $(s_t, a_t, r_t)_{t=1...T}$. The goal of RL is to learn a policy maximizing the sum of rewards during episodes, i.e., the return. In this setting, exploration poses many questions. If the environment provides no rewards, what should the agent look for? When should it act greedily with respect to the rewards it has found and stop looking for more? In the history of RL, many approaches have been proposed to tackle these questions. On one hand, classic single-environment approaches range from intrinsic motivation with visitation counts [2, 4, 13, 26, 53], optimism [1, 5, 25, 27, 31], or curiosity [7, 24, 35, 45, 49, 51], to bootstrapping [12, 33] or empowerment [29, 39]. On the other hand, we find approaches to incrementally learn tasks, such as transfer learning [58], continual learning [28], curriculum learning [32], and meta learning [38]. Below, we review approaches closely related to ours.

---

[2]While it can be argued that the real world has no explicit distinction between training and testing, we use this dichotomy only for the purpose of evaluation.

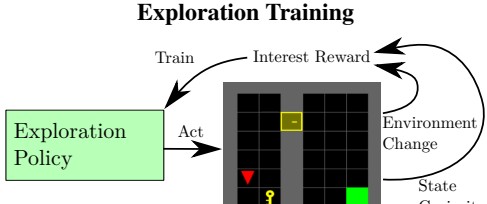

**Exploration Training**      **Exploration Transfer**

Figure 2: **C-BET pre-training.** Our agent interacts with environments and learns using intrinsic rewards computed from state and change counts.

Figure 3: **C-BET transfer.** The pre-trained exploration policy is fixed and guides task-specific policy learning in new environments.

**Intrinsic motivation.**  Exploration strategies relying on intrinsic rewards date back to Schmidhuber [47], who proposed to encourage exploration by visiting hard-to-predict states. More recently, the idea of auxiliary rewards to make up for the lack of external rewards has been extensively studied in RL, supported by evidence from psychology and neuroscience [20]. Several intrinsic rewards have been proposed, ranging from visitation count bonuses [4, 53] to bonuses based on prediction error of some quantity. For example, the agent may learn a dynamics model and try to predict the next state [24, 35, 48, 51]. By giving a bonus proportional to the prediction error, the agent is incentivized to explore unpredictable states. Schultheis et al. [49], instead, proposed to learn intrinsic rewards function by maximizing extrinsic rewards by meta-gradient.

However, in these approaches exploration is agent-centric, i.e., based on an agent's belief such as the forward model error. In contrast, with this work we propose additionally learning *environment-centric* exploration policies. C-BET neither requires a model nor knowledge of extrinsic rewards. Instead, it encourages the agent to perform actions causing *interesting changes* to the environment. We should note that while Raileanu and Rocktäschel [36] proposed a similar approach, their exploration policy lacks the transfer component and also requires to learn models.

**Transfer learning.**  The idea of agents capable of incrementally learning tasks is well-known in the field of machine learning, with the first approaches dating back to the 90s' [40, 41, 56]. In RL, recent methods have focused on policy and feature transfer. In the former, a pre-trained agent (teacher) is used to transfer behaviors to a new agent (student). Examples include policy distillation, where the student is trained to minimize the Kullback-Leibler divergence to the teacher [44] or to multiple teachers at the same time [55]. Alternative approaches, instead, directly reuse policies from source tasks to build the student policy [3, 17, 21]. In feature transfer, a pre-learned state representation is used to encourage exploration when tasks are presented to the agent [22, 59]. Similar to transfer RL, continual RL studies how learning on one or more tasks can help accelerate learning on different tasks, and how to prevent catastrophic forgetting [28, 42, 50]. Meta RL, instead, tries to exploit underlying common structures between tasks to learn new tasks more quickly [18, 38].

However, the setup in these approaches is not task-agnostic, i.e., task-specific policies are transferred rather than exploration policies. For example, after learning a policy maximizing the rewards of one task, the agent starts exploring guided by the same policy as a second task is given. Transfer is task-centric rather than task-agnostic and environment-centric. Consequently, if tasks are too dissimilar information cannot be reused, even if the environments are similar. By contrast, in this work we propose learning task-agnostic exploration from one or many environments and show transfer to unseen environments. We should note that while Pathak et al. [35] did demonstrate fine-tuning on different maze maps, their focus and large-scale evaluations remain on tabula-rasa exploration.

## 3 Learning to Explore

Our goal is to decouple the environment-centric nature of exploration from its agent-centric component. Contrary to prior work, we propose to first learn an environment-centric exploration policy and then to transfer it to unseen environments. The policy is driven by the inherent interestingness of states and is learned over time via interaction. First, during a pre-training phase, the agent interacts with many environments without any tasks and learns an exploration policy. Then, when new environments and tasks are presented, the agent uses the previously learned policy to explore more efficiently and learn task-specific policies. C-BET's key components are (1) a novel intrinsic reward and the learning of a policy to disentangle exploration from exploitation, and (2) the use of such policy to help exploration for new tasks. Figures 2 and 3 summarize our framework.

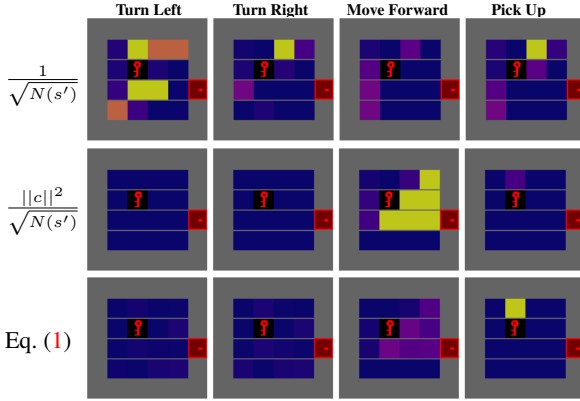

**Turn Left   Turn Right   Move Forward   Pick Up**

$\frac{1}{\sqrt{N(s')}}$

$\frac{||c||^2}{\sqrt{N(s')}}$

Eq. (1)

Gridworld with a key and a door. Observations encode cells depending on their content (e.g., 5 for the key, 10 for the agent). In each cell the agent is facing downward, and can pick up the key only from the cell above it. Samples have been collected randomly.

Figure 4: **Visualization of intrinsic rewards (row) for the agent's actions (column).** Brighter color denotes higher reward. Rewarding only state counts (top) does not provide useful feedback, and going to the corners is valued more than picking up the key. With the L2 norm of state changes (middle), the agent is biased in favor of moving, because its position is encoded with the highest value in the observation space. The resulting policy will prefer to navigate without picking up the key. By contrast, C-BET (bottom) gives picking up the key the highest reward.

We should note that Rajendran et al. [37] also proposed a transfer framework based on intrinsic rewards. In their work, the agent switches between practice episodes –where the agent receives only intrinsic rewards– and match episodes –giving only extrinsic rewards. However, practice episodes are simpler variations of match episodes (e.g., in Atari Pong the agent practices against itself) rather than different tasks as in C-BET. Furthermore, the intrinsic reward used in practice episodes is given by a function trained with meta-gradients to improve the extrinsic-reward return. That is, exploration is not task-agnostic as in C-BET, and extrinsic rewards are the main drive of the agent.

### 3.1 Interestingness of State-Action Pairs

The natural world is filled with states or scenarios that are inherently interesting and our goal is to capture this inherent interestingness via intrinsic rewards. In this paper, we propose adding an environment-centric component of interestingness to the existing agent-centric component of surprise. Specifically, we hypothesize that the environment can *change* on interaction, and the changes that are *rare* are inherently interesting. That is, we penalize actions not affecting the environment, and favor actions producing rare changes. For instance, moving around, bumping into walls, or trying to open locked doors without keys all result in no change and thus will be of low interest.

We also want to keep the agent-centric component in exploration –that is, the exploration policy should look for surprises or unseen states. Thus, we further reward actions leading to less-visited states. By combining these two components, the resulting C-BET interest-based reward is

$$r_i(s, a, s') = 1/(N(s') + N(c)), \tag{1}$$

where $c(s, s')$ defines the *environment change* of a transition $(s, a, s')$, and $N$ denotes (pseudo)counts of changes and states. Figure 4 empirically shows its effectiveness. In Section 4 we discuss change encodings used in our experiments.

### 3.2 Exploration Learning

In this phase, we want to learn task-agnostic exploration policies from interaction with many environments. The agent has no goal, but states where it can 'die' are still terminal. In this setting, we would like to treat the problem of learning exploration as an MDP with intrinsic-rewards only, and train the agent to maximize discounted intrinsic-returns averaged over episodes.

Formally, the agent explores many environments $\mathcal{E}_{\text{EXP}} = \{E_1, E_2, \ldots, E_N\}$, each governed by MDP $\langle S_n, A, P, r_i, \gamma_i \rangle$. That is, each environment has its own states but the action space is the same, and all environments obey the same dynamics $P$ and the same intrinsic reward function $r_i$. The agent's goal is to learn an exploration policy maximizing the sum of discounted intrinsic rewards, i.e., $\pi_{\text{EXP}}(s, a) = \arg\max_\pi \mathbb{E}_{\mathcal{E}, \pi}[\sum_t \gamma_i^t r_i(s_t, a_t)]$. To approximate the average, after a maximum number of steps the environment is reset and a new episode starts, as typically done in RL.

However, both common [7, 35, 36] and Eq. (1) intrinsic rewards decrease over time as the agent explores, to the point that they vanish to zero given enough samples. For instance, counts will grow to infinity, or prediction models error will go to zero. While this is not an issue in the tabula-rasa setup where the agent also gets extrinsic rewards, it can be problematic in the proposed task-agnostic exploration framework. Any policy, indeed, would be optimal if all rewards are zero.

To prevent Eq. (1) from vanishing, we randomly reset counts any given time step. To explain why resets need to be random, we start by considering 'episodic counts' proposed by Raileanu and Rocktäschel [36]. These counts are reset at the beginning of every episode to ensure that the agent does not go back and forth between a sequence of states with high rewards. While this work fine when extrinsic rewards are also given, it can be a problem if we learn only on intrinsic rewards. When counts are reset, the agent 'forgets' past trajectories and thinks that every state and change is new. If resets always happen at the end of an episode, then initial states will always get higher reward. Moreover, starting always with zero-counts may favor some trajectories and penalize others.

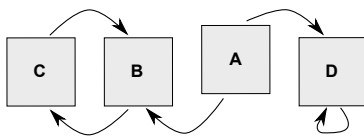

Figure 5: This chainworld illustrates that if counts are resets at the beginning of every episode, the learned policy will never visit D.

Consider for example the chainworld in Figure 5. The agent always starts in A, from where it can go to B or D. From B, it loops between B and C. From D, it cannot go anywhere else. The optimal exploration policy should visit all states uniformly, by randomly going to B and D. However, if we reset counts at every episode the agent forgets that it has already visited B and C. Thus, the intrinsic rewards for B and C are high again, and trajectory ABCBCBC... gives higher intrinsic return than ADDD... Consequently, the optimal policy with respect to episodic counts will always prefer to visit B rather than D.

The optimal exploration policy, instead, should have some randomness to visit the environment uniformly, while prioritizing interesting states. For this reason, we propose to reset counts at any given step with probability $p$. When a new episode starts, counts may not be reset yet so the agent remembers what it has visited before. As the agent explores, on average common states and changes will have higher count more often, and the agent will correctly prefer rarer ones. In this paper, we propose $p \le 1 - \gamma_i$ where $\gamma_i$ is the intrinsic reward discount factor. This is a fitting choice because in an MDP the sum of discounted rewards can be interpreted as the expected sum of undiscounted rewards if every time step had a $1 - \gamma_i$ probability of ending. Intuitively, this means that $\gamma_i$ implies a 'life expectancy' of $1/(1 - \gamma_i)$ steps, and thus resets should not happen more frequently than that.

The resulting MDP with Eq. (1) rewards and random count resets can be solved by any RL algorithm. However, we should note that this MDP is non-stationary, because the agent may receive different rewards for the same state, depending on how many times the state has been visited in the past. Nonetheless, classic intrinsic rewards –even in tabula-rasa exploration– either based on prediction errors [35, 36] or counts [4] also introduce non-stationarity because they change over time as well. In practice, this non-stationarity is not an issue because intrinsic rewards change slowly over time.

### 3.3 Exploration Transfer

Now, the agent is presented with new environments and asked to solve tasks. Formally, each environment is governed by the standard MDP $\langle S, A, P, r, \gamma \rangle$ and the agent's goal is to learn a policy maximizing the sum of extrinsic rewards, i.e., $\pi_{\text{TASK}}(s, a) = \arg\max_\pi \mathbb{E}_\pi[\sum_t \gamma^t r(s_t, a_t)]$. Note that while during pre-training the policy was learned across all environments (one exploration policy for all environments), at transfer we learn one task-specific policy for each environment.

In this phase, the interest-policy learned earlier drives exploration as tasks and environments are presented to the agent. In order not to forget interestingness over time, the exploration policy is added as a fixed bias to the task-specific policy, similarly to what Hailu and Sommer [21] proposed. Thanks to the decoupling of the interest-policy (based on the intrinsic reward) from the task-policy (based on the extrinsic reward), the latter can be also learned independently via any RL algorithm.

In our experiments, we use IMPALA [16] to learn both $\pi_{\text{EXP}}$ and $\pi_{\text{TASK}}$. IMPALA learns policies of the form $\pi(s, a) = \sigma(f(s, a))$, where $\sigma$ is the softmax function. The policy is trained to maximize a function representing the value of states $V(s)$, trained on the given rewards. In our framework, we combine the two policies as follows.

- During pre-training, by using intrinsic rewards we learn $V_i(s)$ and $\pi_{\text{EXP}}(s, a) = \sigma(f_i(s, a))$.
- At transfer, we learn $V_e(s)$ on extrinsic rewards. The policy is $\pi_{\text{TASK}}(s, a) = \sigma(f_e(s, a) + f_i(s, a))$. The interestingness $f_i$ is transferred but not trained, i.e., it acts as fixed bias to encourage interaction. Initially the policy follows $f_i$ since $f_e$ is initialized randomly. As it finds extrinsic rewards, the sum $f_e + f_i$ becomes greedier with respect to extrinsic rewards, and $f_e$ slowly overtakes $f_i$[3].

---

[3]If exploration and the task goals are misaligned, we can decay exploration, e.g., $\pi_{\text{TASK}}(s, a) = \sigma(\alpha f_i(s, a) + f_e(s, a))$, where $\alpha$ decays over time, similarly to common $\epsilon$-greedy policies.

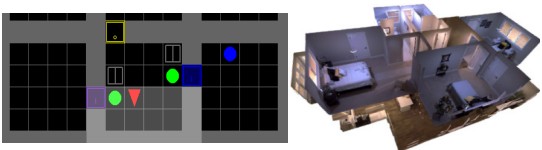 Figure 6: **Examples of the environments used in our experiments.** In MiniGrid (left), the agent navigates through a grid and interacts with objects (keys, doors, boxes, and balls) to fulfill a task. In Habitat (right), the agent navigates through visually realistic rooms.

Note that we transfer only $f_i$ (the policy) and not $V_i$ (the state value). We could think of transferring $V_i$ as fixed bias as well, i.e., by having $V_e(s) = V(s) + V_i(s)$. The policy would be trained on $V_e$ –the states value with respect to the given task– where $V_i$ is fixed and only $V$ is updated. However, we believe it is more beneficial to isolate the exploratory component within the policy, in order to keep the task-specific value function targeted on extrinsic rewards. By not transferring $V_i$, $V_e$ can be accurately trained on extrinsic rewards –that the agent will see often thanks to $f_i$ from the pre-trained policy. $V_e$, in turn, can make $\pi_{\text{TASK}}$ greedy with respect to extrinsic rewards as $V_e$ is learned.

# 4   Experiments

Our experiments are designed to highlight the benefits of disentangling the environment-centric nature of exploration from agent-centric behavior by learning a separate exploration policy and then transferring it to new environments. We stress that for learning task-agnostic exploration there are no standard benchmark environments, experimental setups, well-defined evaluation metrics, or even baselines to compare against. One of our contributions is to provide an exhaustive evaluation framework for the transfer exploration paradigm.

**Environments.** The experiments are divided into two main sections. The first is about MiniGrid [10] (Section 4.1), a set of procedurally-generated environments where the agent can interact with many objects. The second is about Habitat [46] (Section 4.2), a navigation simulator showcasing the generality of our MiniGrid experiments to a visually realistic domain.

**Change encoding.** In both MiniGrid and Habitat the agent partially observes the environment, since it cannot see through corners, closed door, or inside boxes, and has a limited field of view. Rather than egocentric views (i.e., what the agent sees in front of itself), we use $360°$ panoramic views to count environment changes, as this is a rotation-invariant representation of the observed state. Similar to Chaplot et al. [8], we concatenate four egocentric views taken from $0°$, $90°$, $180°$, and $270°$ with respect to the North. Then, the change of a transition is the difference between panoramic views $\mathsf{pano}(s)$, i.e., $c(s, s') := \mathsf{pano}(s') - \mathsf{pano}(s)$.

**Baselines.** We evaluate against the following algorithms. For more details, refer to Appendix A.1.
- *Count* [4]. The intrinsic reward is inversely proportional to the next state visitation count.
- *Random Network Distillation (RND)* [7]. The intrinsic reward is the prediction error of states' random features between a trained network and a fixed one. This can be interpreted as similar to using state counts because the prediction improves states are seen more often.
- *Rewarding Impact-Driven Exploration (RIDE)* [36]. The intrinsic reward is the prediction error between consecutive embedded states, normalized by episodic state counts.
- *Curiosity* [35]. The intrinsic reward is the prediction error between consecutive states.

In Appendix B we investigate the importance of count resets, panoramic changes, and different count-based rewards. The source code is available at https://github.com/sparisi/cbet/.

## 4.1   MiniGrid Experiments

MiniGrid environments [10] are procedurally-generated gridworlds where the agent can interact with objects, such as keys, doors, and boxes (Figure 6). Exploration is challenging because rewards are sparse, observations are partial, and specific actions are needed to visit all states (e.g., pickup key to open door). With MiniGrid, we can generate several pairs of train and test environments that are related but still different in many ways. These pairs enable evaluation of both the learning and transfer abilities of an exploration method and its ability to deal with unseen components.

**Implementation details.** All environments give a $7\times7\times3$ partial observation encoding the content of the $7\times7$ tiles in front of the agent (including the agent's tile). The agent cannot see through walls, closed doors, or inside boxes. The action space is discrete: left, right, forward, pick up, drop, toggle, and done. For a complete description of the environments, we refer to Appendix A.4.

**Setups.** We present three setups, to study different exploration transfers against tabula-rasa.

- *MultiEnv (many-to-many transfer).* The agent loops over three environments episode by episode, and learns the exploration policy using intrinsic rewards only. There is one state count and one change count for all three environments rather than separate counts for each. The environments are: KeyCorridorS3R3, BlockedUnlockPickup, and MultiRoom-N4-S5, and have been chosen for size and interaction variety: the first has both a locked and an unlocked door, a key, and a ball; the second adds a box; the third has more rooms. Note that even if these environments have all object types, the agent cannot experience all kinds of interactions. For example, it will not know that keys can be hidden in boxes, as in the ObstructedMazes. The policy is then transferred to ten environments, seven of which are new. A good intrinsic reward should help learn better exploration faster from multiple environments, thanks to sharing experience from diverse interaction.
- *SingleEnv (one-to-many transfer).* The policy is pre-trained on a single environment. DoorKey and KeyCorridor are used for pre-training because they have some –but not all– objects.
- *Tabula-rasa (no pre-training / transfer).* A task-specific policy is learned as in classic intrinsic motivation by summing intrinsic and extrinsic rewards. While it is a non-realistic setup, it is the most common RL exploration approach, and thus serves as baseline against our transfer framework.

**Evaluation metrics.** Our goal is to learn exploration policies that encourage interaction with the environments and transfer well to new environments, i.e., that can further be trained to solve extrinsic tasks faster. Therefore, we evaluate policies according to the following criteria.

- Unique interactions over 100 episodes at transfer to new environments, after intrinsic-reward pre-training (no extrinsic-reward training yet). Unique interactions are picks/drops/toggles resulting in new environment changes. For instance, repeatedly picking and dropping the same key in the same cell results in only two interactions.
- Tasks success rate over 100 episodes at transfer to new environments, after intrinsic-reward pre-training (no extrinsic-reward training yet). The task success rate denotes in how many episodes the exploration policy visits goal states –thus, would have already solved the environment task.
- Extrinsic return during extrinsic-reward training, after intrinsic-reward training.

### 4.1.1 MiniGrid Pre-Training Results

Figure 7 shows results after pre-training in MultiEnv. In Appendix C we report results for the two SingleEnv setups as well. C-BET policy both interacts with the environment and find goal states more often than all baselines. As we will see in the next section, this will result in faster extrinsic-reward learning. Furthermore, C-BET's policy transfers well to all environments, even the ones with unknown dynamics (e.g., boxes in ObstructedMazes needs to be toggled to reveal keys). Of the baselines, only Count scores high average interactions and success rate, but it does not generalize as well as C-BET. Indeed, most of Count's success comes from environments visited at pre-training (the first five), but most of its interactions are in environments with unseen dynamics (ObstructedMazes). That is, Count's policy can explore familiar environments prioritizing state coverage (high success rate and few interactions), but not unfamiliar ones (low success rate yet high interactions).

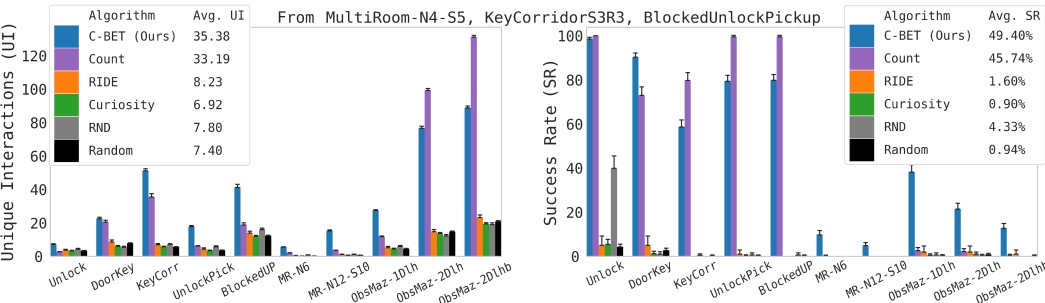

Figure 7: **Unique interactions and success rate** at the beginning of transfer of policies pre-trained in MultiEnv. Not only C-BET interacts the most and achieves the highest success rate, but also interacts and succeeds in **all** environments. Naturally, it interacts more in environment with many keys/balls/boxes to pick (KeyCorridor, BlockedUnblockPickup, ObstructedMazes), and less if there is nothing to pick (MultiRooms). On the contrary, Count overfits to the training environments and performs well only on the first five. Other baselines perform poorly, almost as a random policy.

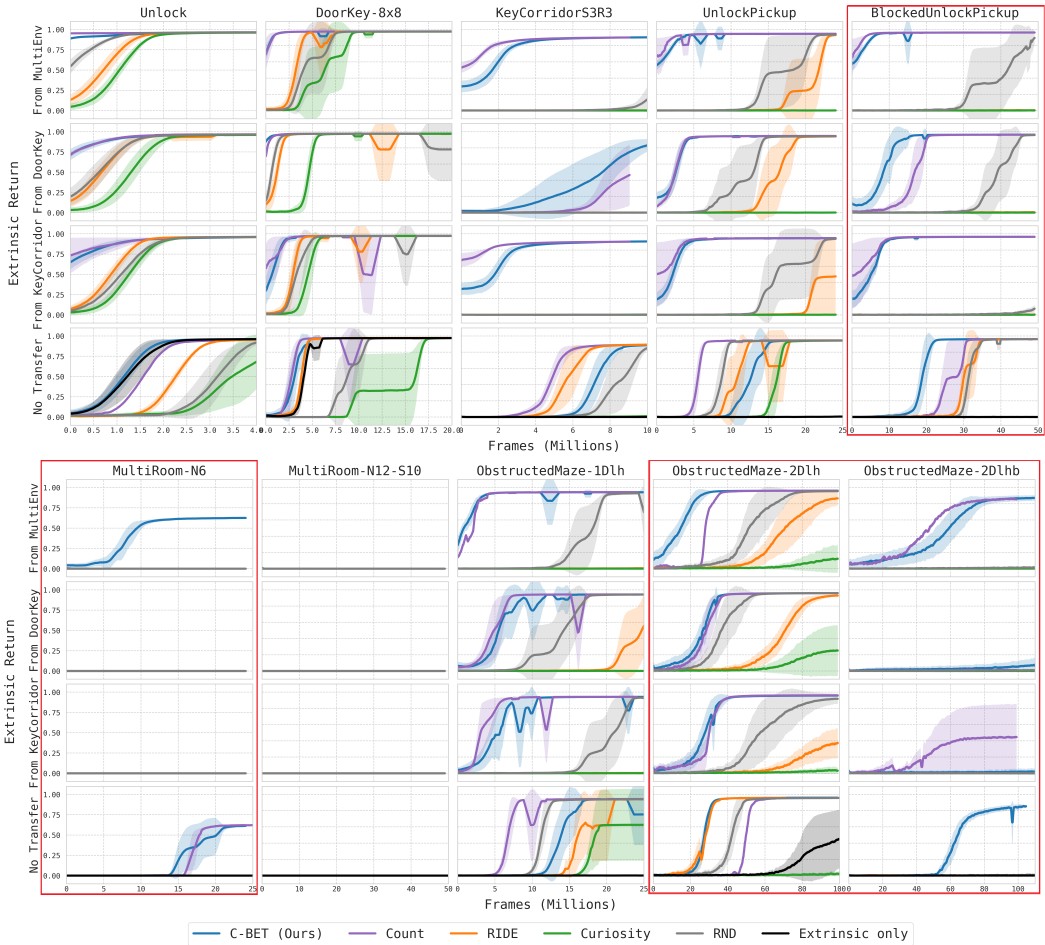

Figure 8: **MiniGrid task learning, for both transfer and tabula-rasa exploration.** The hardest tasks are outlined in red. C-BET (blue) from MultiEnv (top row under each environment) performs the best, starting with nearly optimal policies in most environments. This demonstrates the effectiveness of pre-training on multiple environments using the C-BET intrinsic reward.

Finally, RIDE, Curiosity, and RND baselines perform poorly. This is unsurprising if we consider that they rely on predictive models and that MiniGrid dynamics are deterministic and simple. Dynamics and embeddings models are learned quickly, without giving the policy time to explore. In Appendix C.3 we support this claim by showing the intrinsic reward trend during pre-training.

### 4.1.2  MiniGrid Transfer Results

We transfer the exploration policies learned in Figure 7 as discussed in Section 3.3. Figure 8 shows how transfer setups (many-to-many and one-to-many) perform against tabula-rasa exploration.
The first takeaway is that policies pre-trained with the C-BET intrinsic reward outperform baselines in both transfer and tabula-rasa. In MultiEnv transfer, C-BET performs the best, especially on the hardest environments (outlined in red). In particular, only C-BET is able to transfer to MultiRoom-N6. On the contary, Count –that can solve it in tabula-rasa– fails at transfer. C-BET is also the only solving ObstructedMaze-2Dlhb –the hardest environment among the ten– even in tabula-rasa.
The second takeaway is that baselines relying on models are not suited to the transfer framework. RIDE, Curiosity and RND perform better in the tabula-rasa setup (last row), except for the easiest environments (Unlock and DoorKey), meaning that transfer is actually harmful. These results are in line with Figure 7, where only C-BET and Count show success at offline transfer. Furthermore, RIDE, Curiosity and RND perform worst when transfer is from MultiEnv, highlighting that their intrinsic rewards are not suited for a multi-environment setup.
Finally, no algorithm learns MultiRoom-N12-10, not even C-BET despite showing some success in Figure 7. This is due to the randomly-initialized $f_e$ of the task-specific policy, hindering the pre-trained exploration policy success.

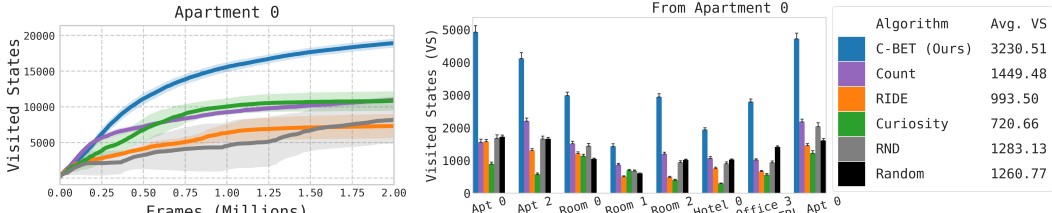

Figure 9: **Habitat pre-training.** C-BET explores the scene faster and scores the highest unique state count.

Figure 10: **Habitat offline transfer.** Bars denote the unique state count in an new scene during one episode. C-BET visits more than twice as many states than all baselines.

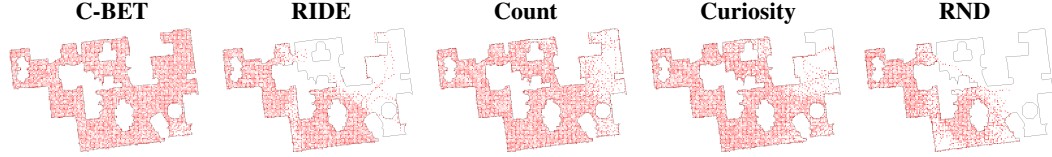

Figure 11: **Scene coverage** of exploration policies during pre-training (2M steps) in Apartment 0. Darker red cells denote higher visitation rates. Only C-BET visits all of the scene uniformly.

## 4.2 Habitat Experiments

To demonstrate that C-BET's efficacy extends to realistic settings with visual inputs, we perform experiments on Habitat [46] with Replica scenes [52].

**Implementation details.** Egocentric views have resolution $64 \times 64 \times 3$. The action space is discrete: forward 0.25 meter, turn $10°$ left, and turn $10°$ right. To ease computational demands, we use #Exploration [54] with static hashing to map both egocentric and panoramic views to hash codes and count their occurrences with a hash table. More details in Appendix A.6.

**Setups.** We evaluate Habitat on the *one-to-many transfer*. First, we pre-train exploration policies with only intrinsic rewards in one scene. Then, we evaluate them on new scenes without further learning. Given a fixed amount of steps, better policies will visit more of the new scenes.

**Evaluation metrics.** Unlike MiniGrid, we use no extrinsic rewards in Habitat. Since the agent has to navigate through rooms and spaces, we evaluate exploration policies using scene coverage measured by the agent's true state in Cartesian coordinates (not accessible by the agent)[4]. Faster, larger and more uniform coverage corresponds to better exploration. Plots show mean and confidence interval over seven random seeds per method with no smoothing.

### 4.2.1 Habitat Pre-Training Results

We pre-train exploration policies on Apartment 0 (Figure 6), one of the largest Replica scene in the dataset. Figures 9 and 11 show state coverage throughout and at the end of pre-training, respectively. C-BET explores more efficiently, covering twice as much of the scene than all baselines. In particular, at the end of pre-training it has explored almost all Apartment 0 uniformly. In Appendix E.1 we also report C-BET results when environment changes are encoded with egocentric views rather than panoramic views.

### 4.2.2 Habitat Transfer Results

Here, we evaluate scene coverage of pre-trained policies in seven unseen scenes for episodes of fixed steps. A better exploration policy will exhibit generalization by covering a larger portion of all scenes as evenly as possible, an impressive feat given the visual complexity of the observations. Indeed, generalization is harder than MiniGrid because the lighting, colors, objects, and layout can be very different between scenes (see Figure 14 in the Appendix). Figures 10 and 12 show that, once again, C-BET clearly outperforms all baselines. Its exploration policy transfer well to all scenes, as it uniformly discovers more states. No baseline comes closer to its results. Actually, in many scenes baselines perform worse than a random policy.

---

[4]To ease memory usage, we round states to 0.05 precision, e.g., 1.26 is rounded to 1.25, and 1.28 to 1.30.

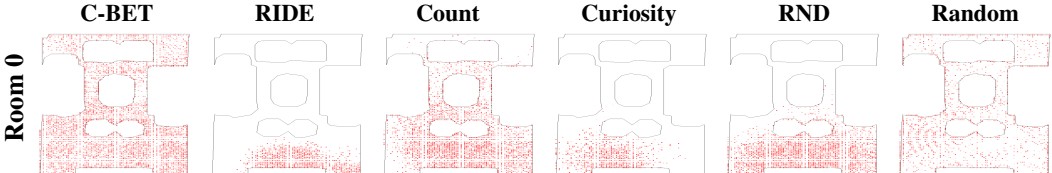

Figure 12: **Scene coverage** of exploration policies after 100 episodes (50,000 total steps) at offline transfer to Room 0. C-BET outperforms baselines and exhibits great transfer by visiting all of the scene uniformly. In Appendix E.2, we show heatmaps for all transfer scenes.

## 5 Discussion

In this paper, we proposed a paradigm change in task-agnostic exploration. Instead of studying task-agnostic exploration in isolated environments, we proposed to (1) learn task-agnostic exploration policies from one or multiple environments, and (2) transfer learned exploration policies to unseen environments at testing time. In our setup, the agent interacts with the environment without any extrinsic goal and learns to explore environments in a task-agnostic manner. To this end, we proposed a novel intrinsic reward to encourage interaction with the environment and the visitation of unseen states. Subsequently, our agent effectively transfers its exploration policy to unseen environments.

**Advantages.** The proposed two-phase framework achieves two important features, making it fundamentally different from prior work. First, we account for *environment interestingness* without relying on additional models. Instead, we use a data-driven approach, estimating the rarity of states and environment changes. Rare changes are considered more interesting, actions causing them receive higher intrinsic rewards, and the agent is encouraged to perform them again. For instance, when navigating through rooms, opening doors will be more interesting due to rarity: the agent must navigate to the corresponding key, collect it, navigate to the door, and finally open it. Thus opening a door is rarer than picking up a key, in turn rarer than simple navigation movements. Furthermore, relying on environment-centric intrinsic rewards rather than task-centric extrinsic rewards facilitates learning from multiple environments at the same time.

Second, contrary to prior transfer and continual learning algorithms we transfer policies learned on *interestingness of the environment* rather than task-specific policies. In the interest-based pre-training phase, we learn through interaction with the environment in a task-agnostic fashion, i.e., the agent freely explores the environment without any extrinsic task.

**Limitations.** In this paper, we assumed that interacting with the environment while looking for rare changes helps find better extrinsic rewards faster. However, exploration and the task goals may be misaligned, thus a highly exploratory policy may slow down the discovery of extrinsic rewards. For instance, the environment may have dangerous states or harmful objects that the agent should avoid, even though they would make it curious during pre-training. Furthermore, C-BET is currently tied to (pseudo)counts to compute the rarity of states and changes. While extensions to continuous spaces exist, count-based metrics are more suited for discrete spaces.

**Impact.** RL can positively impact real-world problems, e.g., healthcare [19], assistive robotics [15], and climate change [43]. Yet, RL may have negative impacts, e.g., in autonomous weapons or workforce displacement [6]. Our work focuses on exploration in RL. Better understanding of what is interesting to do or visit helps exploration in unseen environments, as the agent will not waste time with random actions. Similarly, transferring policies learned in a related setting –as we do– can help narrow the range of the agent's expected behavior. Conversely, in many real-world scenarios exploration by curiosity and interestingness is unacceptable. For instance, autonomous cars cannot run over pedestrians just for the sake of curiosity. At present, our work is far from these impacts, but we hope to direct research to focus more on learning from multiple environments and transferring experiences, while at the same time ensuring the safety and reliability of autonomous agents.

## Acknowledgments and Disclosure of Funding

The authors would like to thank Davide Tateo for his thoughtful discussions, Roberta Raileanu for providing support for RIDE's codebase, and Sudeep Dasari for helping run experiments. VD and DP were supported in part by NSF Fellowship and DARPA Machine Common Sense grant, respectively.

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
