# A  Supplemental Details

## A.1  Algorithms Details

Here, we formally define all intrinsic rewards evaluated in the paper. Let's denote by $N(s)$ the state visitation (pseudo)count, by $N(c)$ the state-change visitation (pseudo)count, and by $\phi(s)$ some state embeddings parameterized by a neural network. Formally, the rewards $r_i(s, a, s')$ we evaluate are:

- **C-BET (Ours)**: $r_i = 1/(N(c) + N(s'))$. At pre-training, both counts are reset with probability $p \leq 1 - \gamma_i$ at every step.
- **Count**: $r_i = 1/\sqrt{N(s')}$. $N(s')$ is never reset.
- **RND**: $r_i = ||\phi(s') - \hat{\phi}(s')||^2$, where $\hat{\phi}$ is a fixed random network, and $\phi$ is trained to minimize the same error.
- **Curiosity**: $r_i = ||f(\phi(s), a) - \phi(s')||^2$, where $\phi$ is trained to minimize the prediction error of both an inverse and a forward model, and $f$ is the forward model.
- **RIDE**: $r_i = ||\phi(s) - \phi(s')||^2/\sqrt{N(s')}$, where $\phi$ is trained to minimize the prediction error of both an inverse and a forward model. $N(s')$ is reset at the beginning of every episode during both pre-training and transfer.

## A.2  Hyperparameter Details

Experiments are built on the codebase developed by Raileanu and Rocktäschel [36], which includes the Torchbeast implementation of IMPALA [30]. All algorithms use the same network architectures.

- **Policy and value function**. The input is the environment partial observation, which is $7{\times}7{\times}3$ for MiniGrid and $64{\times}64{\times}3$ for Habitat. It is passed through three (for MiniGrid) or five (for Habitat) convolutional layers with 32 filters each, kernel size $3{\times}3$, stride 2, and padding 1. An exponential linear unit [11] is after each convolution layer. The output of the last convolution layer is fed into two layers with ReLU activation and 1,024 units, and then into an LSTM [23] with 1,024 units. Finally, two separate fully connected layers of 1,024 units each are after the LSTM, and are used for the value function and the policy, respectively.
- **State embeddings**. The partial observation is passed through five convolutional layers with 32 filters each, kernel size $3{\times}3$, stride 2, and padding 1.
- **Inverse model**. The input is the concatenation of two successive state embeddings. It is fed into two layers with ReLU activation and 256 units.
- **Forward model**. The input is the concatenation of the state embedding and the action. It is fed into two layers with ReLU activation and 256 and 128 units, respectively.

Observations and changes counts are based on egocentric and panoramic views, respectively. For MiniGrid, egocentric views are 147-dimensional while panoramic views are 588-dimensional.
For Habitat, views are encoded using HashSim [9, 54] to ease computational demands. Habitat views are 12,288- and 49,152-dimensional and are both hashed with 128 bits.

Value functions and policies are updated every 100 steps with IMPALA [16], using mini-batches of 32 samples and the RMSProp optimizer [57] (learning rate 0.0001 linearly decaying to 0, momentum 0, and $\varepsilon = 0.00001$). Gradients are clipped to have maximum norm 40.

Both the intrinsic and extrinsic rewards discount factors are 0.99 as goals can be reached in less than 100 steps. This is also the default discount factor used by Raileanu and Rocktäschel [36]. The counts reset probability is $p = 0.001$, meaning that the agent has an approximate 'life expectancy' of 1,000 steps. This is a fitting choice because the largest time horizon for the environments is 640 steps (see Appendix A.4), and $p \leq 1 - \gamma_i$.

Intrinsic rewards and losses are further scaled down by a coefficient for numerical stability.

- Intrinsic reward: 0.1 (RIDE, RND, Curiosity), 0.005 (C-BET, Count).
- Policy entropy loss (all algorithms): 0.0005.
- Value function loss (all algorithms): 0.5.
- Forward model loss (RIDE, Curiosity): 10.
- Inverse model loss (RIDE, Curiosity): 0.1.
- Random network loss (RND): 0.1.

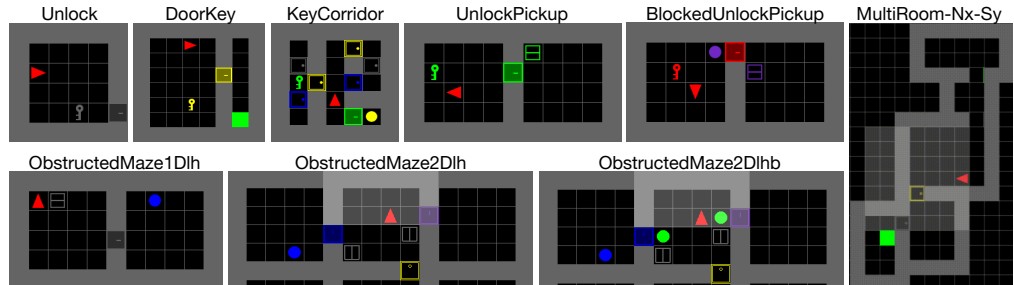

Figure 13: **The MiniGrid environments.** The agent has to navigate through a grid and interact with different objects (keys, doors, boxes, balls) to fulfil a task. At each episode, the grids are procedurally generated, changing rooms layout, objects positioning and color.

## A.3 Compute Details

Experiments were run on a SLURM-based cluster, using a NVIDIA Quadro GP100 GPU and 40 CPUs. For MiniGrid, IMPALA uses 40 actors. One pre-training run takes ~8 hours for 50M steps. Learning time after transfer ranges from ~30 minutes (5M steps in *Unlock*), to ~55 hours (200M steps in *ObstructedMaze2Dlhb*). For Habitat, IMPALA uses 4 actors to prevent memory errors. One pre-training run takes ~45 hours for 2M steps (because Habitat simulation is slower, network inputs –observations– are larger, and true state counts are saved for visitation heatmaps).

## A.4 MiniGrid Environments Details

Figure 13 shows the MiniGrid environments used in this paper. Below is a list of their respective tasks (the hardest are bolded). $T$ denotes the maximum number of steps per episode. Everything is as implemented by default in MiniGrid codebase.

- *Unlock*: pick up the key and unlock the door ($T = 288$).
- *DoorKey-8x8*: pick up the key, unlock the door, and go to green goal ($T = 640$).
- *KeyCorridorS3R3*: pick up the key, unlock the door, and pick up the ball (only the door before the ball is locked) ($T = 270$).
- *UnlockPickup*: pick up the key, unlock the door, and open the box ($T = 288$).
- ***BlockedUnlockPickup***: pick up the ball in front of the door, drop it somewhere else, pick up the key, unlock the door, and open the box ($T = 576$).
- *ObstructedMaze1Dlh*: open the box to reveal the key, pick it up, unlock the door, and pick up the ball ($T = 288$).
- *ObstructedMaze2Dlh*: same as above, but with two doors to unlock ($T = 576$).
- ***ObstructedMaze2Dlhb***: same as above, but with two balls in front of the doors (like in BlockedUnblockPickup) ($T = 576$).
- ***MultiRoom-N6***: navigate through six rooms of maximum size ten and go to the green goal (all doors are already unlocked) ($T = 120$).
- ***MultiRoom-N12-S10***: same as above, but with ten rooms of maximum size twelve ($T = 240$).
- *MultiRoom-N4-S5*: smaller MultiRoom environment (five rooms of maximum size five), used only for pre-training ($T = 100$).

In all tasks, the extrinsic reward is $r_t = 1 - 0.9 \ (t/T)$. This reward is given only for solving the task. The action space is discrete with seven actions: left, right, forward, pick up, drop, toggle, and done. 'Toggle' unlocks a door if the agent has the corresponding key, opens/closes a door if unlocked, and open boxes to reveal keys. 'Done' is implemented by default in MiniGrid codebase and it is used for language-based tasks only, thus it does nothing in our tasks.

The grid is procedurally generated at each episode, and the agent's initial position is random within a fixed area far from the goal (e.g., in DoorKey the agent starts in the area with the key, or in MultiRooms it starts in the farthest room from the goal).

BlockedUnblockPickup, ObstructedMaze2Dlh, and ObstructedMaze2Dlhb provide all types of interaction, but only ObstructedMazes have hidden keys. MultiRooms differ from the other environments because they have more rooms, all doors are already unlocked, and the goals are further away from the agent start position. They require more steps to be completed, and this explains their smallest extrinsic return compared to other environments in Figure 8.

## A.5 MiniGrid Plot Smoothing Details

MiniGrid's plots show the mean and confidence interval over seven random seeds per method, smoothed over 300 epochs with a sliding window. Each epoch consists of 3,200 samples (32 minibatches of 100 steps). These samples are used for both updating the agent and computing evaluation metrics. This smoothing is necessary because we need to wait for the end of the episode to compute its return, but episodes are longer than 100 steps (see Appendix A.4). Thus, in some epochs we cannot compute the expected return and we end up with less than one evaluation data point per epoch. On the contrary, for Habitat visitation counts we can always retrieve the number of visited states at every step without waiting for the end of the episode, therefore there is no need for smoothing.

## A.6 Habitat Environments Details

Figure 14 shows the Replica scenes used in this paper. At each step, the agent can move forward by 0.25 meter or turn left/right by $10°$. The episode length is $T = 500$ steps. Egocentric views (used as policy input and to count states) have resolution $64 \times 64 \times 3$. Figure 15 shows the agent's $360°$ panoramic view, used to count environment changes. For computational ease, we use the following static hashing to map both views to hash codes and count their occurrences with a hash table.

$$\mathsf{cbet\_hash}(x) = \begin{cases} 1 & \text{if } g(x) > 0.5 \\ -1 & \text{if } g(x) < -0.5 \\ 0 & \text{otherwise,} \end{cases} \tag{2}$$

where $x$ is either an egocentric or panoramic view, and $g(x) = \tanh(Ax) + w$. $A$ and $w$ are a projection matrix and a bias vector, respectively, both with i.i.d. entries drawn from the standard normal distribution. Our hashing is inspired by SimHash [9] binary encoding $\mathsf{sim\_hash}(x) = \mathsf{sign}(Ag(x))$. In preliminary results, our ternary encoding performed better than SimHash.

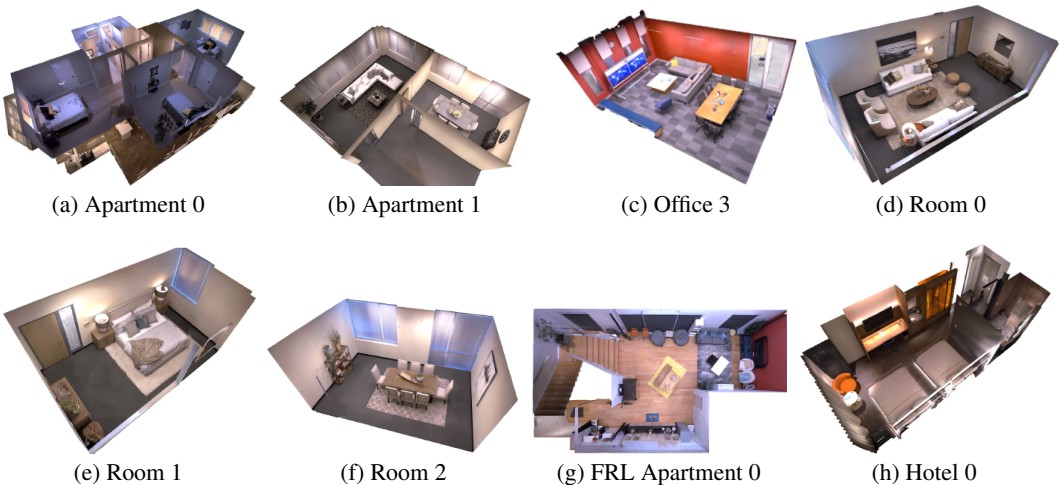

(a) Apartment 0      (b) Apartment 1      (c) Office 3      (d) Room 0

(e) Room 1      (f) Room 2      (g) FRL Apartment 0      (h) Hotel 0

Figure 14: **Real-world scenes from the Replica dataset used for our Habitat experiments.** The scenes have several rooms and obstacles that make exploration challenging. Apartment 0 is used for pre-training, while the remainder are used to evaluate transfer policies.

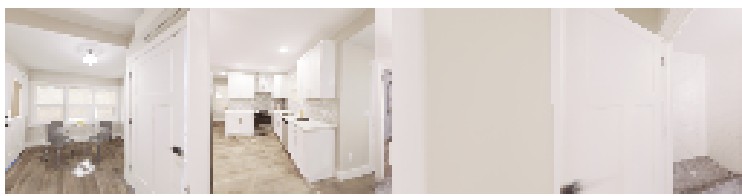

Figure 15: **Agent's $360°$ panoramic view of Apartment 0.** The view concatenates four egocentric images taken from $0°$, $90°$, $180°$, and $270°$ with respect to the North.

# B    Ablation Study on MiniGrid

In this paper, we propose the following C-BET intrinsic reward:

$$r_i(s, a, s') = \frac{1}{\underbrace{N(c)}_{\text{change part}} + \underbrace{N(s')}_{\text{state part}}},$$

where $c(s, s')$ is the environment change of a transition $(s, a, s')$, and $N$ denotes (pseudo)counts of changes and states. Both counts are randomly reset at any given time step, independently from each other. We use this reward for two reasons. First, as shown in Figure 16, summing the two parts **within** the square root yields high rewards **only** to transitions with **both** low state and change counts. Second, the ablation study in Appendix B.2 shows that squaring the reward performs best.

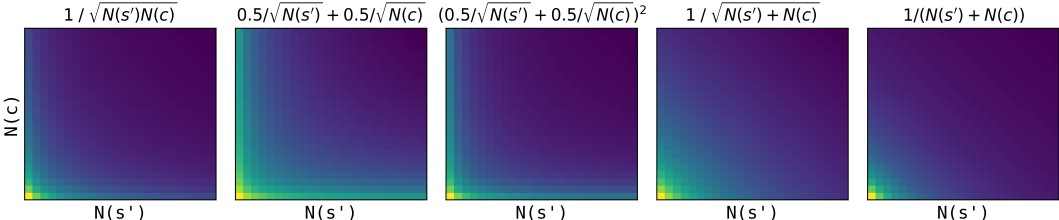

Figure 16: Heatmaps of different intrinsic rewards on varying state and change counts. The brighter the color, the higher the reward. Considering the sum of the parts –either squared or not– (second and third maps) still rewards transitions with either low state or change counts –not necessarily both. This is shown by the bright edges of the heatmap. The same happens with only the product (first map), albeit to a lesser extent. If we sum the parts inside the square root (fourth and fifth map) we correctly reward only transitions with both low state and change counts. Between the latter two, squaring the reward further penalizes transitions with high counts, because the reward decreases faster as either the state or change count increases.

**Aim of the Study**

The key components of C-BET reward are count random resets and the change-based reward. In this section, we investigate these components and answer the following fundamental questions:

- How important are count resets?
- How important is how we encode changes? Are panoramic views crucial for C-BET?
- How do different combinations of the state and change terms in Figure 16 perform? Is Eq. 3 truly the best?
- How do pre-training environments affect the quality of the exploration policy?

We start by investigating if panoramic views are good for counting states, and if squaring the reward can help state-only baseline as well (Appendix B.1). We continue by comparing the different types of rewards shown in Figure 16 (Appendix B.2). Subsequently, we evaluate C-BET with egocentric views for counting changes (Appendix B.3). Then, we further review interesting questions arising from such evaluation regarding change-based rewards (Appendix B.4). Finally, we study the importance of count resets. Finally, we conclude by showing what really matters for C-BET (Appendix B.5).

As evaluation metric, we consider the 'average episode success' of pre-trained policies, in both the *one-to-many* (SingleEnv) and *many-to-many* (MultiEnv) setups discussed in Section 4. After pre-training, policies are transferred to new environments and have to solve their respective task. Without further training, we evaluate their average success rate over 100 episodes. The higher the success rate, the faster the policy is likely to solve the task if further extrinsic-reward training would be carried one, as the extrinsic reward will be seen more often.
We present results through error bar plots and recap tables, reporting mean and confidence interval over seven seeds. Error bar plots show the success rate for each new environment a policy is transferred to, and each setup (SingleEnv and MultiEnv) has a dedicated plot. Figure legends average the rate over the ten environments. Tables further average it over the three setups.

## B.1 State-Only Counts: Egocentric vs Panoramic Views, Squaring vs Classic Reward

**Questions:** Are panoramic views good for counting states? Does squaring the state-only baseline increase its performance?

**Discussion:** Figure 17 shows that egocentric views (plain patch) perform best, with an average success rate of 28.3% over the three setups. Panoramic views (dashed patch) perform well in MultiEnv but poorly in SingleEnvs, for an average success rate of 24.9%. The reason is that panoramic views are rotation-invariant: when the agent turns left/right the panoramic state does not change, the state count increases and the intrinsic reward decreases. In the end, the agent will not be encourage to turn at all. This is not surprising, because panoramic views are designed to represent *environment changes* rather than the agent's state[5]. Only in MultiEnv panoramic views perform well. Thanks to the diversity of visited states, counts do not increase too rapidly, thus turning around is not penalized too much. In Appendix B.4 we further investigate this issue with 'change-only counts' rewards.

Figure 17 also shows that squaring the reward (yellow) performs slightly worse, since policies seem to overfit to the training environments.

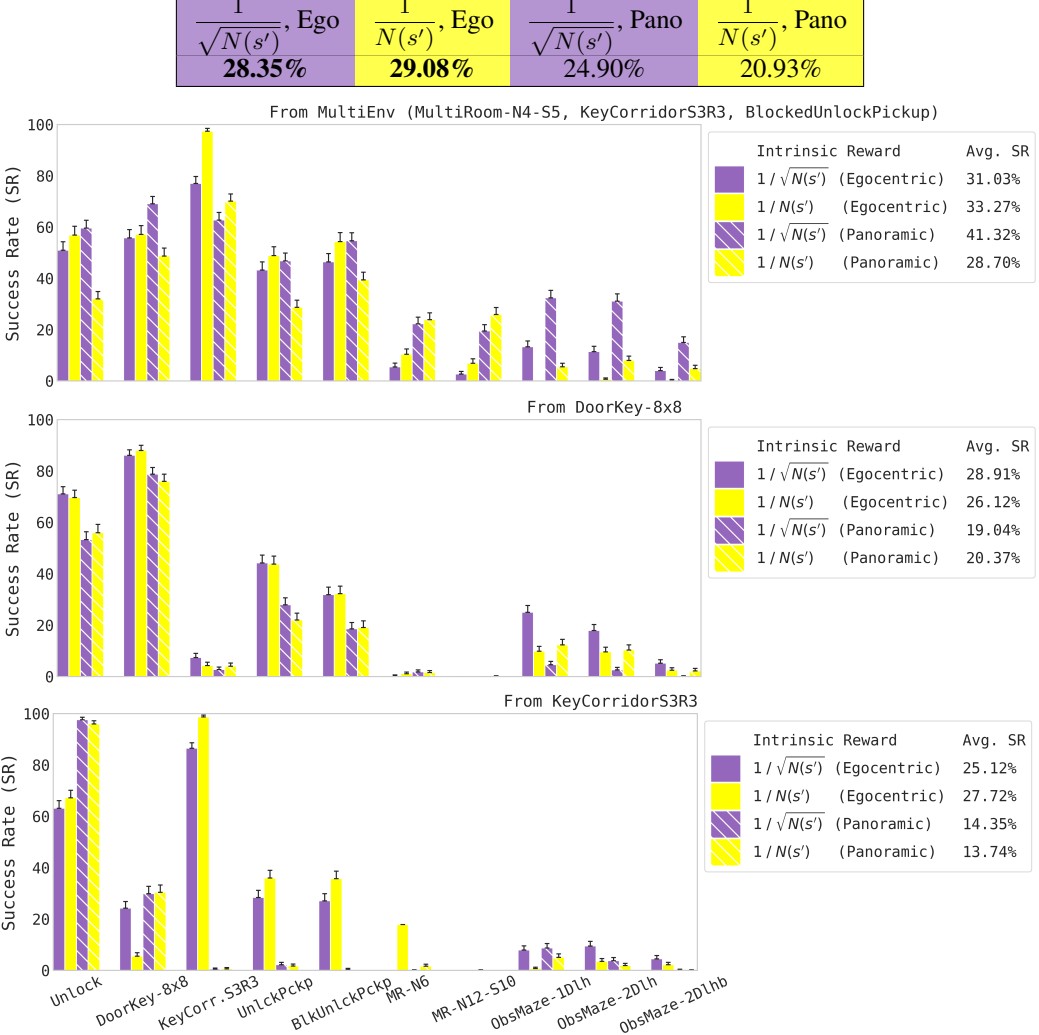

Figure 17: Success rate of policies pre-trained on different state-only intrinsic rewards with count resets. Squared rewards tend to overfit to the training environments. Panoramic views perform worse than egocentric views in SingleEnvs, but better in MultiEnv due to the diversity of visited states.

---

[5]Note that panoramic views are used only for the intrinsic reward. The policy still receives egocentric views.

## B.2 C-BET: Different Reward Combinations

**Question:** How do different combinations of state and change rewards perform? Is Eq. (3) better than other combinations?

**Discussion:** Figures 18 shows that Eq. (3) reward indeed performs best, achieving the highest success rate (blue, 33.64%). Furthermore, it also generalizes to the most environments, as it is the only showing good transfer to ObstructedMazes, especially when pre-trained on MultiEnv. Considering both the sum and the product of the parts (violet, 31.9%) comes in a close second, but does not transfer to ObstructedMazes when pre-trained on MultiEnv. Other reward combinations perform comparably (red, gold, green, 30.4%). Nonetheless, they all perform better than 'state-only count' (purple, 28.35%, Figure 17). Interestingly, 'change-only count' perform poorly (brown, 18.15%). We will come back to this in Appendix B.4.

| $\dfrac{1}{\sqrt{N(s')N(c)}}$ | $\dfrac{0.5}{\sqrt{N(s')}}+\dfrac{0.5}{\sqrt{N(s')}}$ | $\left(\dfrac{0.5}{\sqrt{N(s')}}+\dfrac{0.5}{\sqrt{N(s')}}\right)^2$ | $\dfrac{1}{\sqrt{N(c)+N(s')}}$ | $\dfrac{1}{N(c)+N(s')}$ | $\dfrac{1}{\sqrt{N(c)}}$ |
|:---:|:---:|:---:|:---:|:---:|:---:|
| 30.43% | 30.40% | 31.92% | 30.45% | **33.64%** | 18.15% |

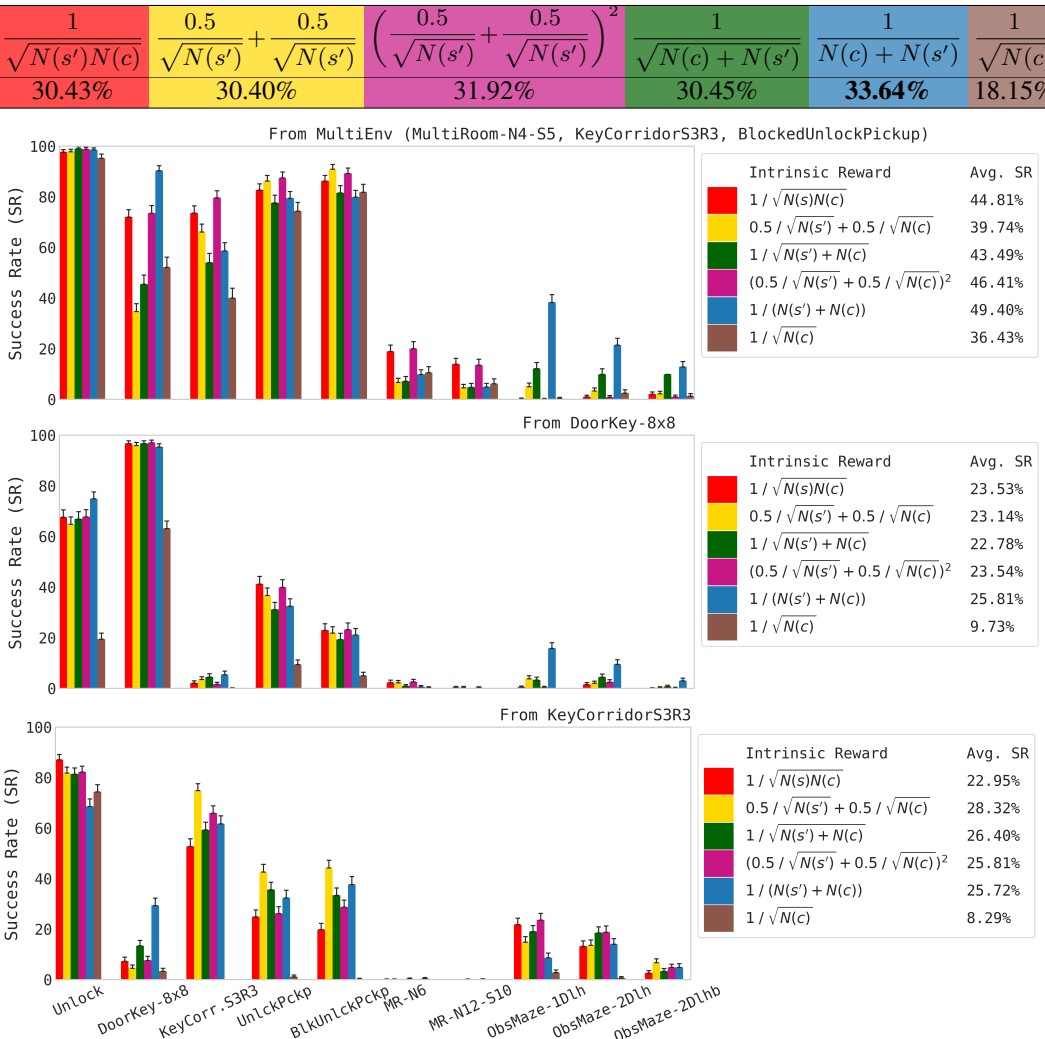

Figure 18: Success rate of policies pre-trained on different combinations of state and change count rewards, with panoramic changes and random resets. States are always counted with egocentric views. C-BET (blue) outperforms other rewards, achieving the highest overall success rate and being the only transferring well to ObstructedMazes.

## B.3 C-BET: Egocentric Views

**Question:** How does C-BET perform when environment changes are encoded with egocentric views?
**Discussion:** With egocentric views, the performance of all rewards decreases. This is not surprising, because panoramic views better encode environment changes, being rotation-invariant w.r.t. the orientation of the agent. The key to better exploration, thus, is the combination of agent-centric encoding of the state (egocentric views) and environment-centric encoding of the change (panoramic views). This conclusion is also strengthened by the following fact: neither 'state-only count' (Figure 17, purple and yellow) nor 'change-only count' (brown) achieve the same success rate and transfer generalization as C-BET (blue). *C-BET does not perform better because of panoramic change counts, but because of how it combines them with egocentric state counts.*

| $\dfrac{1}{\sqrt{N(s')N(c)}}$ | $\dfrac{0.5}{\sqrt{N(s')}} + \dfrac{0.5}{\sqrt{N(s')}}$ | $\left(\dfrac{0.5}{\sqrt{N(s')}} + \dfrac{0.5}{\sqrt{N(s')}}\right)^2$ | $\dfrac{1}{\sqrt{N(c)+N(s')}}$ | $\dfrac{1}{N(c)+N(s')}$ | $\dfrac{1}{\sqrt{N(c)}}$ |
|:---:|:---:|:---:|:---:|:---:|:---:|
| **27.19%** | 19.61% | **26.95%** | 17.33% | 21.69% | 6.64% |

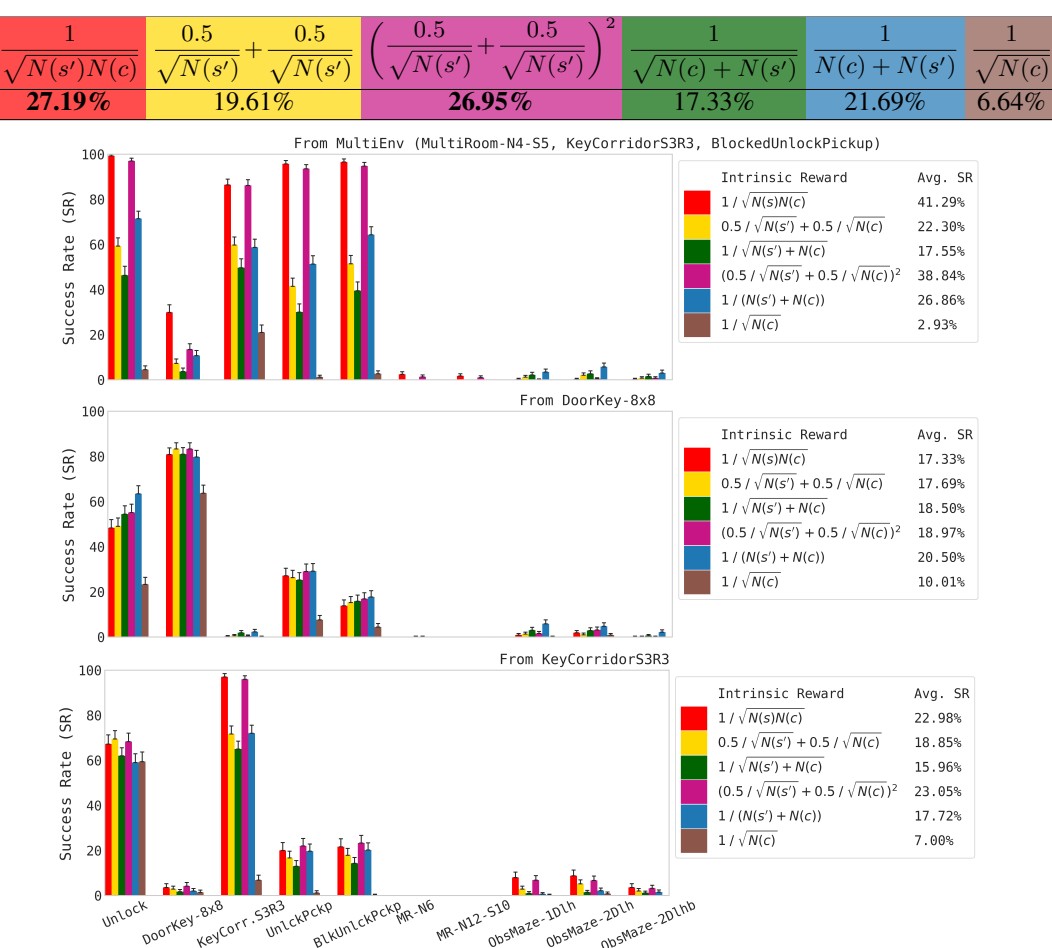

Figure 19: Compared to Figure 18, egocentric changes decrease the performance of all rewards, especially the one based only on change counts (brown). Despite its lower success rate, Eq. (3) still transfer well to many environments, including ObstructedMazes when pre-trained on MultiEnv.

## B.4 C-BET: Why Do 'Change-Only Counts' Perform Poorly?

**Question:** In the previous sections, 'change-only count' (brown) performed poorly compared to other rewards. However, 'state-count only' (purple) does not. Why is that so?
**Discussion:** First, 'panoramic change-only counts' performs poorly only in SingleEnvs and not in MultiEnv. This is similar to what we have seen in Figure 17 with 'state-only counts': in SingleEnvs panoramic change counts for turning left/right increase too fast and rewards decay too rapidly, while in MultiEnv the diversity of states prevents that. Thus, the resulting policy will learn not to turn.

Then why does 'change-only count' perform even worse with egocentric changes, especially in MultiEnv? In this case, the problem is the opposite. Turning left/right is the only action **always** resulting in some egocentric change, while other actions (including moving forward) can fail and thus produce no change. Furthermore, in visually rich environmnents, such as KeyCorridor, the egocentric change is likely to be unique. Therefore, the policy will overfit to turning left/right.

This is confirmed by Figure 20, showing the policy distribution at the end of pre-training. When trained with panoramic changes (second plot), 'change-only counts' overfits to moving forward. When trained with egocentric views (third plot), the policy is drastically different and prefers to turn most of the time. On the contrary, C-BET policy (first plot) moves less and interacts more.

Interestingly, C-BET also does nothing ('done') more than other policies, but this can be explained by recalling what we discussed in Section 3.2: the optimal exploration policy should keep some randomness to visit the environment uniformly. In practice, C-BET does not have to always interact with the environment if the resulting change is not diverse. That is, C-BET intrinsic reward tries to keep uniform state/changes counts over **all** possible state/changes, including 'no change' counts.

Finally, Figure 20 also shows how the action distribution changes depending on the environment. In larger environments (DoorKey, BlockedUnblockPickup, MultiRooms and ObstructedMazes), the agent moves more. In environments where doors are already unlocked (KeyCorridor and MultiRooms) the agent toggles more and picks/drops fewer times.

| | Left | Right | Forward | Pick | Drop | Toggle | Done | Entropy |
|---|---|---|---|---|---|---|---|---|
| C-BET | 10.7% | 11.1% | 51.5% | 7.5% | 7.0% | 9.3% | 2.6% | 0.78 |
| Pano Change-Only | 11.5% | 9.8% | 58.4% | 5.5% | 5.1% | 9.5% | 0.05% | 0.68 |
| Ego Change-Only | 34.9% | 35.0% | 24.7% | 2.0% | 1.9% | 1.2% | 0.1% | 0.66 |

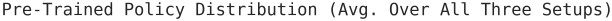

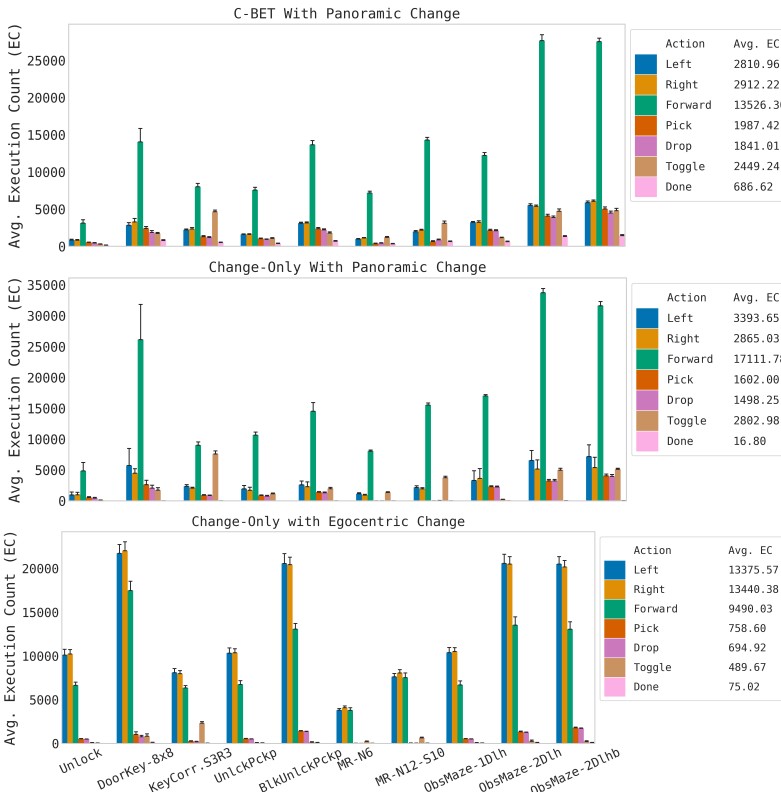

Figure 20: Plots show how many times pre-trained policies executed an action during 100 episodes when transferred to new environments. Counts are averages over both SingleEnvs and MultiEnv transfers. The table reports the resulting action probability. Last column shows the normalized entropy of the distribution (a random distribution has entropy 1). 'Panoramic change-only' overfits to moving forward, while 'egocentric change-only' to turns. On the contrary, C-BET moves less, interacts more, and overfit less to any action as shown by its higher entropy.

## B.5 Counts: No Resets vs Random Resets

**Questions:** Are count random resets necessary for intrinsic-only learning?
**Discussion:** Figure 22 and its table show that not resetting counts achieves lower success rate. In particular, transfer is significantly worse when the agent is pre-trained on DoorKey. The reason for the worse performance –especially in DoorKey– is shown in Figure 21. As the agent explores, intrinsic rewards decay due to counts growth. The decay is more prominent in DoorKey because of its sparse changes and similar states, but it is noticeable also in KeyCorridor and MultiEnv.
Interestingly, 'state-count only' (purple) achieves a better success rate without resets when pre-trained in MultiEnv. However, Figure 22 also shows that the policy overfits to training environments, not showing any transfer to MultiRooms and ObstructedMazes. On the contrary, despite the overall success rate, C-BET (blue) still transfers well to all environments from MultiEnv.

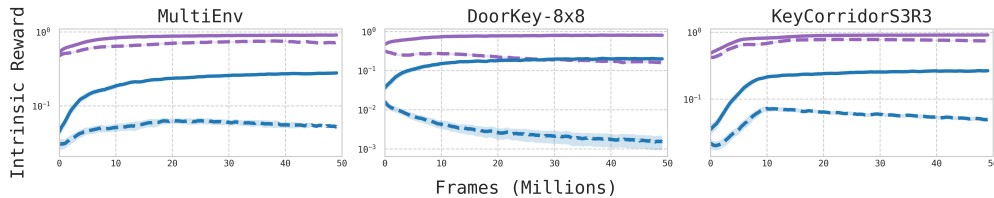

Figure 21: Log scale intrinsic rewards trend at pre-training. Without resets, rewards decay preventing further learning. This is prominent in DoorKey, where states are similar and changes sparse.

| $\dfrac{1}{\sqrt{N(s')}}$, Resets | $\dfrac{1}{N(s') + N(c)}$, Resets | $\dfrac{1}{\sqrt{N(s')}}$, No Resets | $\dfrac{1}{N(s') + N(c)}$, No Resets |
|---|---|---|---|
| 28.35% | **33.67%** | 27.21% | 18.73% |

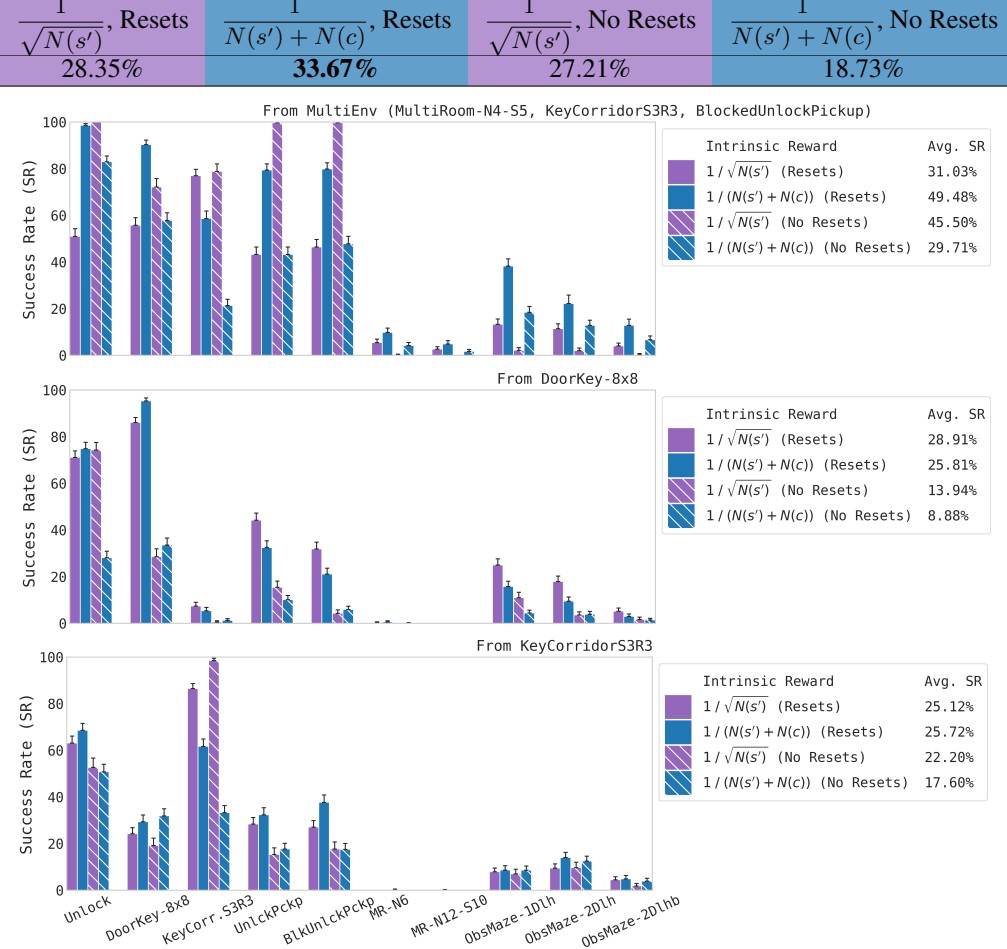

Figure 22: Due to intrinsic rewards decay, policies cannot be properly trained, and do not transfer well. Yet, C-BET still transfers well to all environments when pre-trained in MultiEnv.

## B.6 Conclusion of the Study

The ablation study above has provided the answers to the fundamental questions we asked at the beginning.

1. In Appendix B.1 and B.4, state-only and change-only counts with panoramic views did not perform well, thus *panoramic views, alone, are not sufficient for learning good exploration policies*.
2. In Appendix B.2 and B.3, combining egocentric views for counting states and panoramic views for counting changes performs best, thus *the most important contribution of C-BET is the combination of agent-centric (egocentric states) and environment-centric (panoramic changes) exploration*. The design of the reward is also important, albeit to a lesser extent.
3. In Appendix B.5, resetting counts prevents the vanishing of intrinsic rewards, thus *random resets are also a crucial contribution of C-BET*.
4. In all experiments, policies transfers better when pre-trained in MultiEnv –especially C-BET– thus *pre-training by exploring multiple environments allows better generalization*. Similarly, transfer is harder when pre-training environments have similar states or sparse changes.

# C MiniGrid Pre-Training Supplemental Results

In this paper, we argued that interacting with the environment while looking for rare changes helps finding extrinsic rewards faster. Thus, we evaluated policies on the number of interactions with the environment and on their success rate. Here we further elaborate the results presented in Section 4.1.

## C.1 Unique Interactions per Episode at Pre-Training and at Offline Transfer

Figures 23 and 24 show that C-BET intrinsic reward encourages to interact with the environment more than all baselines. The more C-BET explores at pre-training, the more it interacts with it producing **unique** changes. At transfer, C-BET behavior transfers well to new environments, even the ones unknown dynamics (e.g., boxes in ObstructedMazes needs to be toggled to reveal keys). Of the baselines, only Count performs similarly to C-BET, but it does not generalize as well as C-BET. Most of its interactions, indeed, are in ObstructedMazes after training in MultiEnv.

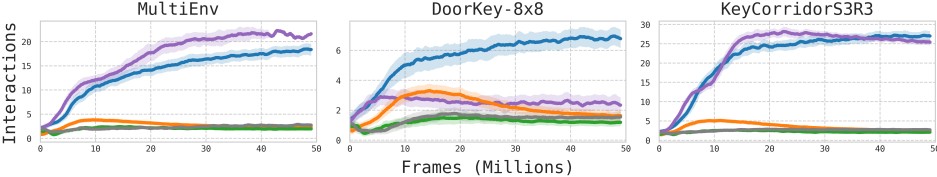

Figure 23: Trend of unique interactions per episode, i.e., picks/drops/toggles resulting in unseen changes. For instance, repeatedly picking and dropping the same key in the same cell results in only two interactions. Only C-BET interacts more with all pre-training environments as it explores.

| C-BET | Count | RIDE | Curiosity | RND | Random |
|-------|-------|------|-----------|-----|--------|
| **26.92** | 24.72 | 8.32 | 7.18 | 7.34 | 7.40 |

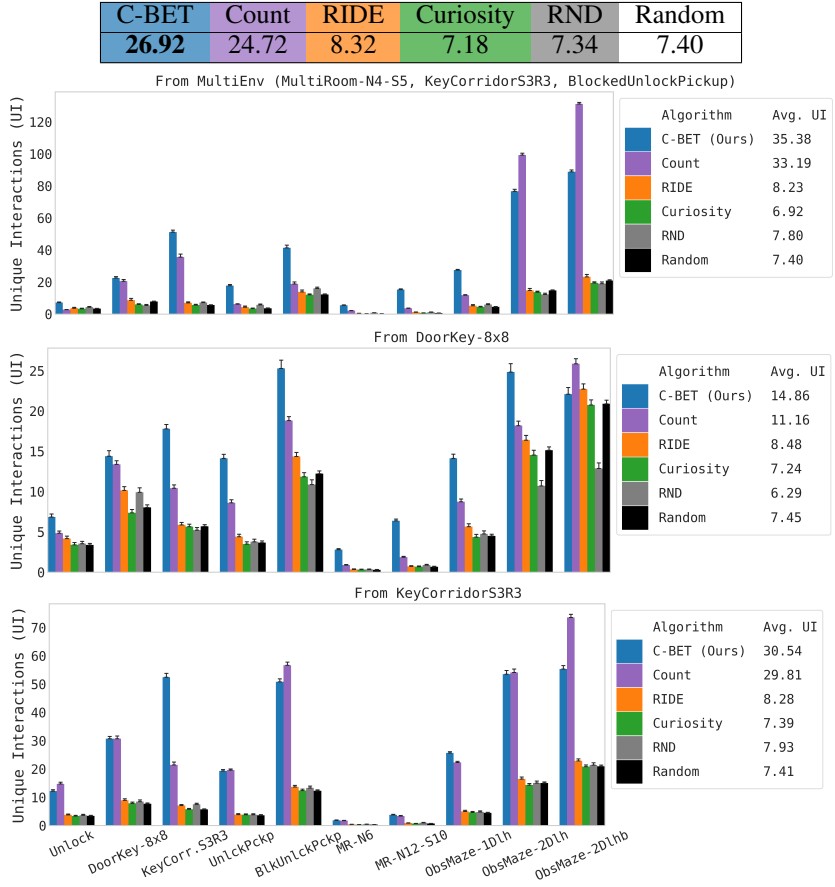

Figure 24: C-BET also outperforms baselines when we compare unique interactions at offline transfer. Not only it interacts the most as shown by the recap table, but also interacts in all environments. Clearly, it interacts more in environment with many keys/balls/boxes to pick (KeyCorridor, BlockedUnblockPickup, ObstructedMazes), and less if there is nothing to pick (MultiRooms).

## C.2 Extrinsic Return at Pre-Training and Success at Offline Transfer

Already at pre-training, C-BET finds goal states thanks to its better exploration, as shown by the increasing trend of the extrinsic return[6] (Figure 25). Similarly, at the beginning of transfer –i.e., without further training with extrinsic rewards– C-BET already succeeds in many environments, especially when pre-trainined on MultiEnv (Figure 26). Among the baselines, only Count achieves success in some environments, but it clearly overfits to the pre-training environments and does not generalize nearly as well as C-BET.

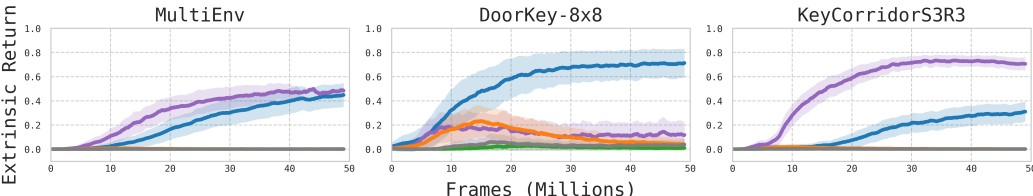

Figure 25: Trend of extrinsic return at pre-training. Only C-BET shows increasing return in all setups, showing that its better exploration brings it often to goal states.

| C-BET | Count | RIDE | Curiosity | RND | Random |
|---|---|---|---|---|---|
| **33.64%** | 27.21% | 1.8% | 0.83% | 2.52% | 0.9% |

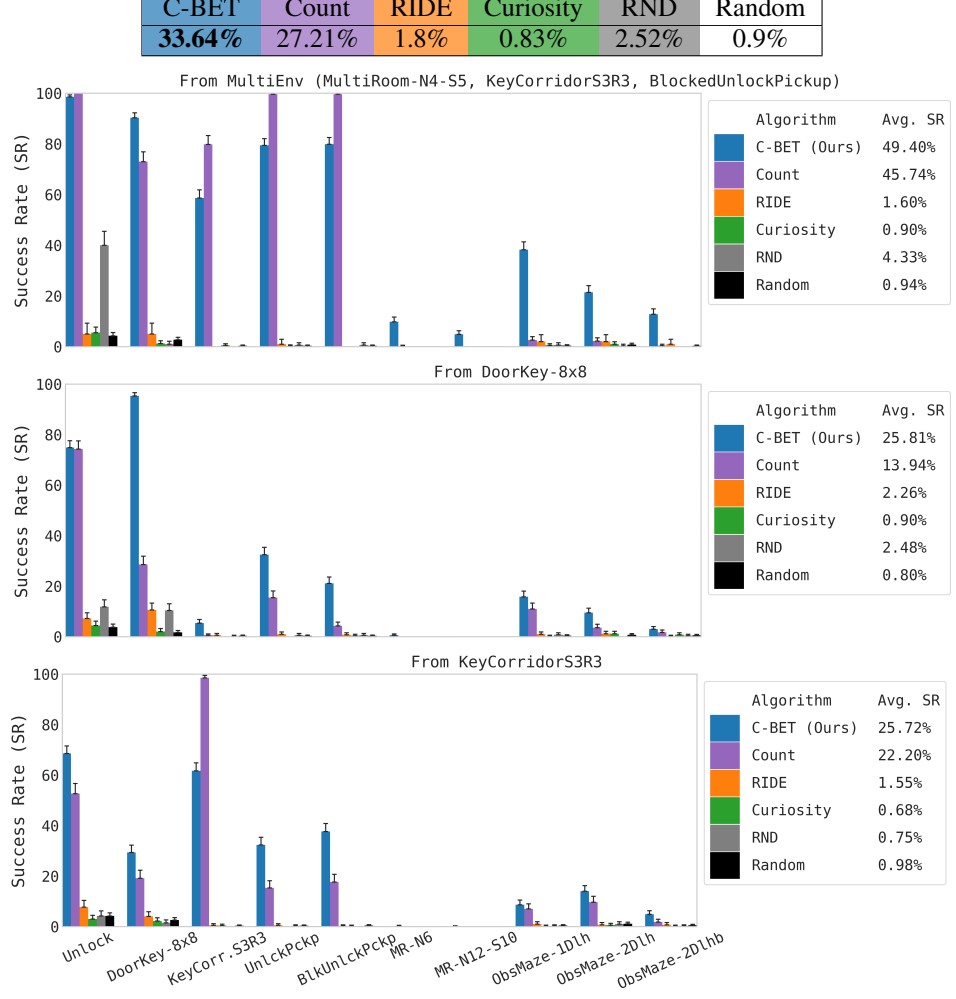

Figure 26: Success rate of pre-trained policies at the beginning of transfer. Not only C-BET achieves the highest rate, but also generalizes to most environments, especially when pre-trained on MultiEnv.

---

[6]We recall that at pre-training agents do not get extrinsic rewards. We only record them as proxy for success.

## C.3  The Problem of Vanishing Intrinsic Rewards at Pre-Training

In the previous two sections, we have shown that C-BET outperforms baselines. Its agent interacts more with the environment while looking for rare and unique changes, and the resulting behavior allows it to discover extrinsic rewards more often. But why do baselines –especially RIDE, Curiosity and RND– perform poorly? As discussed in Section 3.2, classic intrinsic rewards decrease over time as the agent explores, to the point that they vanish to zero given enough samples, preventing further learning. On the contrary, thanks to count resets C-BET reward never decays, and allows the agent to always get meaningful feedback.

This is shown in Figure 27, where intrinsic rewards based on model errors (RIDE, Curiosity, RND) quickly decay as models are learned with ease. Even Count, that does not use any model, suffer from vanishing rewards, albeit to a lesser extent and mostly when trained in environments with similar states like DoorKey. On the contrary, C-BET intrinsic reward increases over time the more agent learns to interact with the environment. But does C-BET outperform Count only thanks to count resets? As we have shown in Appendix B.5, even with resets Count does not generalize well to unseen environments, and overfits to the training environments. C-BET, instead, achieves success even in unseen environments.

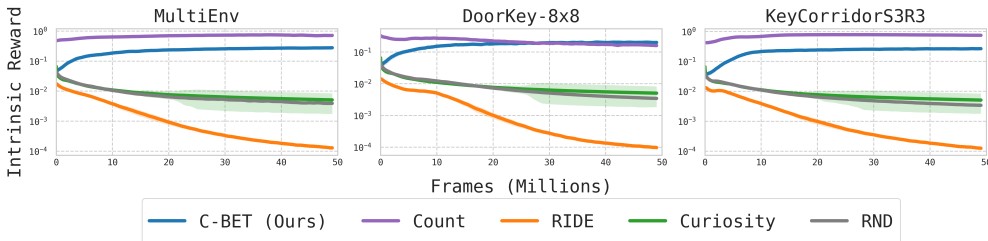

Figure 27: Log-scale trend of intrinsic rewards at pre-training. Baselines rewards decay over time, preventing learning. The problem is prominent in model-based rewards (RIDE, Curiosity, RND). On the contrary, C-BET rewards increase thanks to count resets.

## D  Noisy Environment Supplemental Results

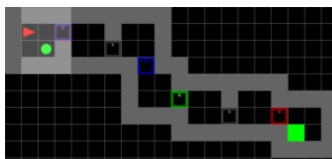

Figure 28: Random instance of the MultiRoomNoisyTV-N7-S4 environment used in our experiments. It has seven rooms of maximum size four.

So far, we conduced experiments in environments with deterministic dynamics. It is known, that stochastic dynamics can be problematic for intrinsic rewards based on model prediction errors [35], as the agents it attracted to sources of noise. This may affect C-BET as well, especially if the noise influences the environment (e.g., the change) rather than agent (e.g., the state). In this section, we test C-BET ability to deal with stochasticity. In particular, we want to see if either its change-based reward or its count resets may negatively affect learning. Therefore, for this evaluation we compare C-BET against the Count baseline –i.e., state-count reward only– with and without count resets on two versions of the same environments (with and without noise)[7]. The environment is MultiRoomNoisyTV, developed by Raileanu and Rocktäschel [36] and depicted in Figure 28. In this environment, a ball is placed in the first room, where the agent spawns at the start of an episode. Anytime the agent does the 'drop' action the ball randomly changes color, regardless of the agent position. Like any MultiRoom, there is nothing that can be picked, and the ball itself cannot be picked.

Table 1 shows that both C-BET and Count perform worse without resets. In particular, Table 2 shows that their learned policies are essentially random. This further confirms what we showed in Appendix B.5: without resets, intrinsic rewards vanish and any policy –even a random one– is optimal. This problem is prominent in MultiRooms due to their scarce state diversity, as there are only empty rooms with already unlocked doors.

---

[7]We also tested RIDE, RND and Curiosity, but they showed the same poor performance of previous experiments. Despite the noise, their model error still decays quickly.

Table 1: **Policy success rate** averaged over all ten transfer environments in different pre-training setups. In both cases, not resetting counts perform worse. However, C-BET is negatively affected by the NoisyTV, albeit to a limited extend, while Count benefits from it.

|  | C-BET (Resets) | C-BET (No Resets) | Count (Resets) | Count (No Resets) |
|---|---|---|---|---|
| Default | 11.37% | 1.68% | 9.72% | 5.92% |
| NoisyTV | 9.76% | 4.0% | **17.90**% | 4.04% |

Unsurprisingly, when the noisy ball is added C-BET performance slightly decreases. However, Count's almost doubles. To explain these behaviors, we need to look at the policy distributions after pre-training, shown in Table 1. Let's consider distributions after training without noise first. In this case, both policies assign high probability to 'toggle'. 'Toggle', indeed, is the only interesting action to do because there is nothing to pick and all doors are already unlocked (and 'toggle' opens/closes doors). The only significant difference is that C-BET moves more and turns less. The reason is that C-BET's panoramic changes give little reward to turns, as discussed in Appendix B.1 and B.4. Overall, when trained without noise C-BET transfer slightly better than Count, as shown in Table 1.

Let's now consider distributions after training with noise. As expected, both policies assign much higher probability to 'drop', the action randomly changing the ball color. Yet, other actions probabilities are significantly different between C-BET and Count.
Indeed, C-BET assigns slightly lower probability to all other actions, but overall its distribution does not change much. This explains why its success rate in Table 1 is similar with/without noise.
On the contrary, Count's 'toggle' probability drastically decreases (from 24.7% to 6.8%) but 'forward' increases. Therefore, Count's policy toggles less and moves more. Thanks to this, the resulting policy explores better, despite the higher 'drop' probability.

Finally, we can summarize the findings of this investigation as follows.
- Count resets are important for learning exploration policies, especially if the training environment lacks diversity.
- A policy 'toggling too much' performs poorly.
- In the presence of noise, C-BET performs worse, albeit to a limited extent.
- The presence of noise actually helps the Count baseline over 'overfit less' to some actions (in this case, the policy learns not to 'toggle too much').

Table 2: **Policy distributions** after pre-training in different setups. Without resets (gold rows) both C-BET and Count policies are essentially random regardless of the noise, as shown by their entropy. With resets and no noise, both C-BET and Count assign low probability to 'drop' and high to 'toggle' (highlighted in red), as there are only unlocked doors. With resets and noise, both assign much higher probability to 'drop', the action triggering the noisy ball to change color.

**From Default MultiRoom**

|  | Left | Right | Forward | Pick | **Drop** | **Toggle** | Done | Entropy |
|---|---|---|---|---|---|---|---|---|
| C-BET (Resets) | 4.0% | 4.3% | 48.6% | 5.3% | **6.0**% | **26.1**% | 5.6% | 0.75 |
| C-BET (No Resets) | 13.6% | 13.6% | 16.0% | 14.1% | 14.1% | 14.4% | 14.0% | 0.99 |
| Count (Resets) | 12.4% | 9.2% | 33.2% | 5.2% | **5.6**% | **29.0**% | 5.0% | 0.86 |
| Count (No Resets) | 14.3% | 14.8% | 23.7% | 12.7% | 10.6% | 12.6% | 11.1% | 0.98 |

**From MultiRoom with NoisyTV**

|  | Left | Right | Forward | Pick | **Drop** | **Toggle** | Done | Entropy |
|---|---|---|---|---|---|---|---|---|
| C-BET (Resets) | 2.9% | 3.0% | 48.4% | 5.3% | **12.8**% | **24.7**% | 2.9% | 0.74 |
| C-BET (No Resets) | 11.5% | 11.3% | 20.3% | 13.0% | 15.8% | 15.7% | 12.3% | 0.99 |
| Count (Resets) | 10.7% | 11.6% | 42.3% | 4.6% | **22.2**% | **6.8**% | 2.1% | 0.82 |
| Count (No Resets) | 10.9% | 12.1% | 16.5% | 9.3% | 32.1% | 11.3% | 7.6% | 0.94 |

# E   Habitat Supplemental Results

## E.1   C-BET With Egocentric Change

Here, we report C-BET results when environment changes are encoded with egocentric views rather than panoramic views.

Figure 29 shows that egocentric views do not perform as well as panoramic views. Yet, C-BET still outperforms all baselines. Indeed, it visits more of Apartment 0 (at pre-training) and of all scenes (at transfer) than all baselines. This is a further confirmation of what stated in Appendix B.3: *C-BET does not perform better because of panoramic change counts, but because of how it combines them with egocentric state counts.* Taking into account **both** agent-centric and environment-centric components is the key for better exploration. Nonetheless, these results also suggest that how we encode *what changes in the environment* as opposed to the agent's state can further improve exploration.

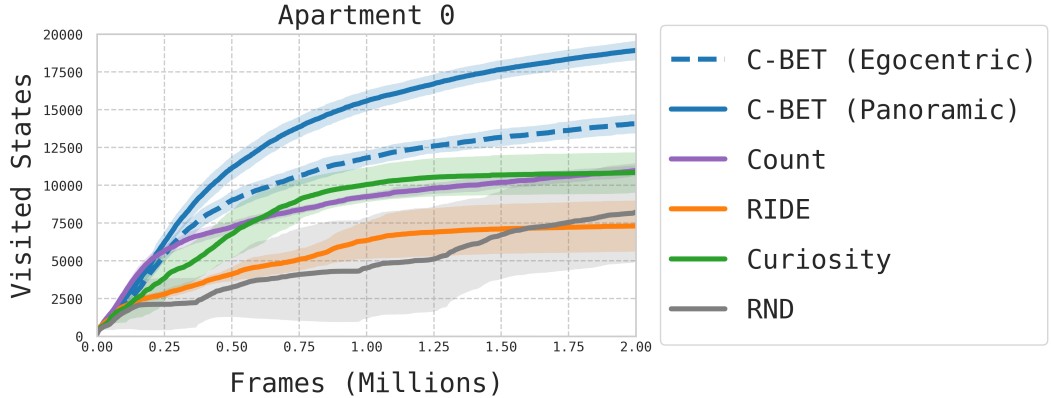

(a) Visited states throughout pre-training.

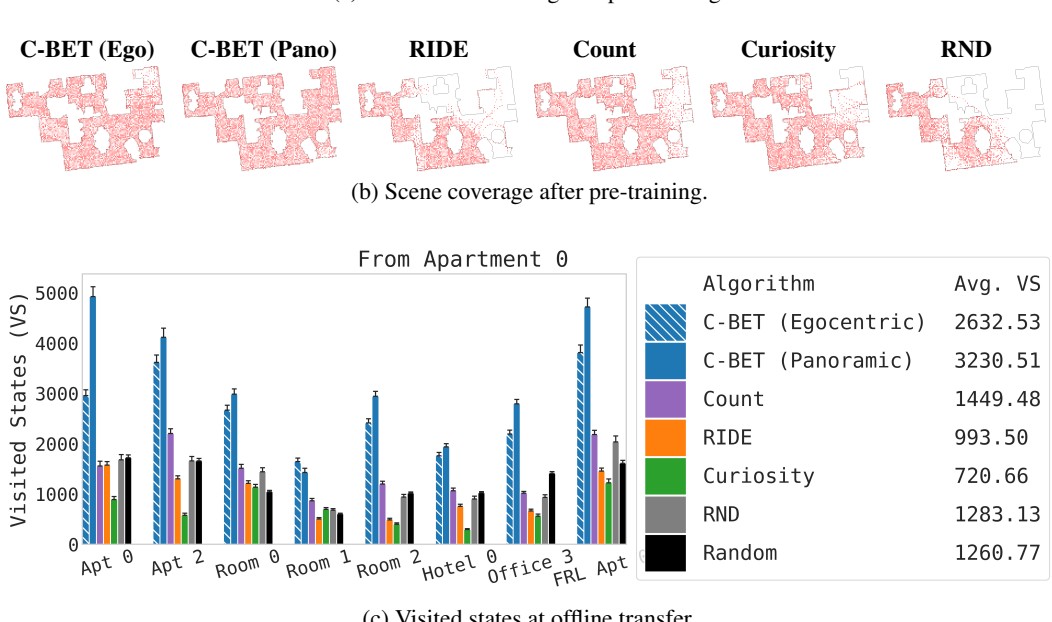

(b) Scene coverage after pre-training.

(c) Visited states at offline transfer.

Figure 29: **Habitat pre-training.** Even with egocentric views, C-BET clearly outperforms all baselines. Not only it still visits almost all Apartment 0 during pre-training, but its policy also transfers well to all unseen scenes.

## E.2 Scene Coverage at Transfer on All Scenes

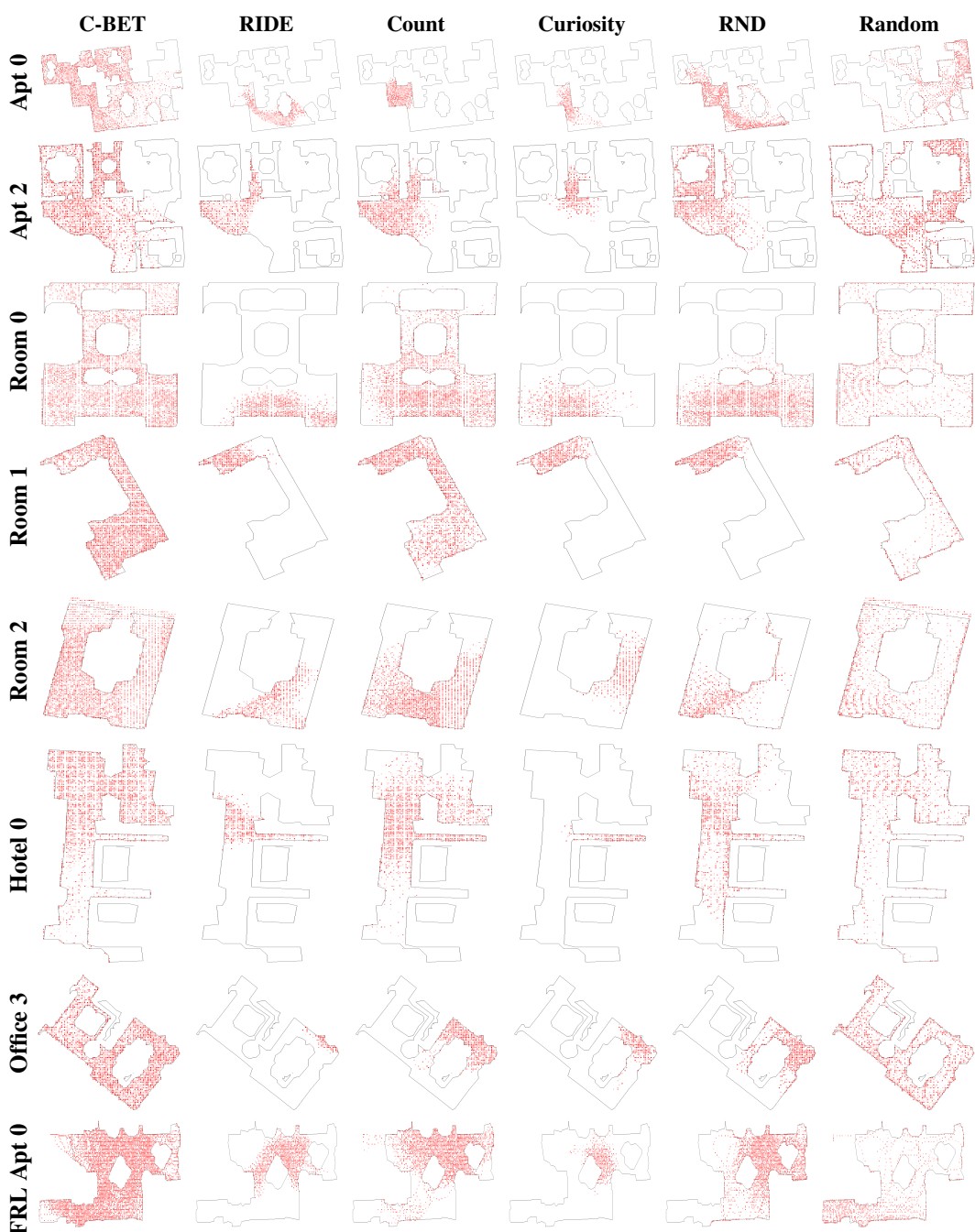

Figure 30: **Scene coverage** of exploration policies at offline transfer to new scenes. Each scene is explored for 100 episodes (50,000 total steps), with the agent always starting at the same location. Darker red cells denote higher visitation rates. C-BET outperforms baselines and exhibits great transfer to all scenes. It is the only algorithm visiting smaller scenes completely (Room 0, Room 1, Room 2, Office 3, FRL Apt. 0), and most of the bigger ones (Apt. 0, Apt. 2, Hotel 0) within only one episode. Furthermore, its visitation count is uniform in all scenes.