# OpenReview forum: "Interesting Object, Curious Agent: Learning Task-Agnostic Exploration"
_NeurIPS.cc/2021/Conference — NeurIPS 2021 Oral_

### Official Review · Reviewer_gCxP · 2021-06-29

**Rating:** 8
**Confidence:** 4

**Summary:**

This paper introduces a novel problem setting within the subfield of exploration in reinforcement learning, and a novel method aimed at solving this problem setting. Specifically, in the new problem setting the agent first gathers experience in several environment with no access to extrinsic reward. The agent is then transferred to similar environments where it must perform the task at hand. What makes this interesting from an exploration perspective is the fact that all the environments are hard exploration problems, so one possible good approach is to learn a good exploration policy in the initial environments, such that when the agent transfers to the new environments it can quickly optimise the extrinsic reward. The new method (CBET) is based on combining standard state-visitation counts with a novel state-change-count, and episode resetting of the counts. The method gets better performance than existing methods (which are targeted at the tabula rasa exploration setting) in terms of exploration in the initial environments and speed of learning in the new environments, with experiments on MiniGrid and Habitat.

**Limitations And Societal Impact:**

Comments on limiations have been adaquately addressed in the paper, with any additional concerns raised in the review above.

**Main Review:**

## Overview
Overall, I believe the paper is just about worthy of acceptance. I think the novel problem setting of transfer-exploration is a useful problem setting that hasn't been studied before, and is more relavant to real-world settings than pure tabula rasa exploration. The benchmarks provided for this problem setting aren't the most thorough or rigorous (another benchmark in which transfer-performance can be measured on a down-stream task would be beneficial), but do still match the capability the problem setting aims to capture. The extensive experiments show that existing methods focused on tabular rasa exploration fail to perform as well in this new setting. The novel method (CBET) proposed for this setting contains some interesting insights about the difference between this setting and tabula rasa exploration, and shows better performance in this setting. I believe the paper could be improved with more experiments ablating different parts of CBET, to see which components are most useful for learning an exploration policy which transfers well to down-stream tasks. Further, a variety of clarifications are needed to ensure that the methodology described in the paper is clear and unambiguous.

## Strengths

As described above, I think the novel problem is interesting and useful, and more relevant to real-world problems than tabular-rasa exploration. The proposed benchmarks adaquately assess the capabilities required in the problem setting. The thorough experiments on existing methods shows that they fail in this problem setting (which isn't surprising, but is useful information), and further experiments in the appendix suggest why this is (due to model learning and hence intrinsic reward collapsing relatively early for these methods). Their novel method is well motivated to solve these problems, and does demonstrate better performance in that benchmarks that existing methods. In general the paper is well written and easy to read.

## Weaknesses and suggestions for improvements
There are several ways in which the paper isn't as strong as it could be

  - While the problem setting is defined informally, it would be useful to have a formal definition, which corresponding terminology. This would enable discussion of the various parts of the problem setting more easily, and make it clear exactly the kind of problem you're trying to present and subsequently solve.
  - While CBET is demonstrated to perform better in the benchmark, and some analysis of the method of resetting is done in the appendix, I think further analysis, and specifically ablations, would be useful to understand why CBET performs better than a simple count-based method. Specifically, the two adjustments CBET makes to a count-based method is the (multiplicative) inclusion of the state-change-count as part of the intrinsic reward, and the use of count resetting after every episode during pretraining. Ablations which remove these two components would be useful to see which of them contribute to performance, and in what way.
  - While I don't think it's necessary for acceptance, another benchmark which had a down-stream extrinsic reward to optimise for (or adjusting the Habitat benchmark to have a downstream task which the policy is transferred to) would improve the robustness of the results and ensure that the capability being measured matches the capability the problem setting desires.
  - Again, I don't believe this is necessary for acceptance, but the current SOTAs on MiniGrid is (I believe) [BeBold: Exploration Beyond the Boundary of Explored Regions](https://arxiv.org/abs/2012.08621) and [Adversarially Guided Actor-Critic](https://arxiv.org/abs/2102.04376), neither of which are compared against in this paper. If these comparisons could be added, it would improve the results in the paper and give evidence that CBET is a valuable new contribution. In particular AGAC doesn't suffer from intrinsic reward disappearing during training, and BeBold uses episode count resets as part of it's intrinsic reward calculation.

### Smaller points

  - Related to Figure 7: It would be beneficial to have an average task-specific reward (or solve rate) for CBET and each of the baselines in MiniGrid, averaged across all transfer tasks, to have a number to compare between the methods. This could be a number related to sample efficiency as well as or instead of task reward.
  - As part of improving our understanding of CBET's performance, It would be great to see the values of the state-count and change-count plotted for different environemnts and different settings. For example, my intuition is that the change counts (and the state counts) for the Habitat environment, given the observation is so rich, should never be the same. Is this the case, or movement in the state-space discrete in this environment, and hence states are often repeated?
  - Related to the above point on BeBold and AGAC, it would be informative to see results of CBET on the standard "No Transfer" setting in MiniGrid across a wider range of environments, such that it can be compared to other exploration methods which are targetted at that setting. While it isn't relevant to the main concern of your paper, it would be informative to see whether CBET's performance on the transfer-exploration setting comes from just better exploration in general, or is specifically most useful in this setting.
  - L23: If the rewards are unknown then the goal isn't defined, so this isn't an entirely realistic setting - there's nothing to achieve. I think you should stick to sparse rewards as a motivating example for exploration.
  - L66: Which approaches? This sentence is unclear to me.
  - L112: "In feature transfer, during..." you've missed a word in this sentence I think.
  - L115: I think it might be worth citing work such as [Decoupling Exploration and Exploitation for Meta-Reinforcement Learning without Sacrifices](https://arxiv.org/abs/2008.02790), which is a meta-RL work which aims to learn an exploration policy which is useful in down-stream tasks, and is hence similar in some sense to your problem setting. In general, I think it could be worth contrasting with this kind of Meta RL setting in the related work setting. In L117 you claim that task-specific policies are transferred, but this generally isn't the case in the Meta-RL setting.
  - L139: "[...]and the changes that are rare": add "that"
  - L158: At this point it would be worth discussing the limitations of count-resets - each episode the agent could explore the same parts of the environment, effectively memorising a policy which hits all the novel states and interesting changes it can, but never exploring the environment beyond that point.
  - L226: "...ten new environments, seven of which are new." - This should be just "...ten environments, seven of which are new..." I think.
  - L240: That's quite a lot of smoothing - are evaluations very high variance? What do less smoothed curves look like?

### Clarification Questions
  - What's the difference between CBET (Ours) and Our Reward in Figure 7?
  - In Figure 7 no-transfer MultiRoom-N12-S10, RIDE gets 0 reward after 100M time-steps, but in the original RIDE paper, RIDE achieves 0.6 extrinsic reward after ~20M time-steps (see Figure 3 (h) in that paper). Do you have an idea why there's that discrepancy in results? Other results from the RIDE paper roughly match the corresponding no-transfer experiments in your paper.
  - In your experiments do you continue training with the intrinsic reward combined with the extrinsic reward after transfering to the new environments, or do you solely train with extrinsic reward after transfer and intrinsic reward before transfer? If you're only training the pre-trained policy with extrinsic reward in the new environment that might explain the discrepancy in scores pointed out in the bullet point above, but I also think that it's not a valid evaluation methodology - there's no reason to disallow intrinsic rewards and further exploration bonuses being used after transfer to a new environment.
  - L178: Your method is unclear to me here: IMPALA learns a state-value function $V(s)$, and a policy $\pi(a|s)$, but doesn't learn a Q-function $Q_i(s,a)$ as you've written it. How do you adapt IMPALA to learn a Q-function?

## Conclusion
I believe the paper is marginally above the acceptance threshold, but I think if the adjustments and additional experiments described above are added (especially the ablations) then I'd be willing to raise my score further.

## Edit:
Given the comments in this response and your response to other authors, I believe my confusions have been clarified, and I believe the paper will be much improved with the improvements you've described here and elsewhere. Hence, I'm raising my score to an 8 - I believe the paper is an interesting, novel and relevant problem setting and presents a novel approach to solving it, while also bein rigorous in the evaluation of their and previous works to ensure that their contribution is robustly useful.


**Time Spent Reviewing:**

3.5

---

> ### Author Response · Authors · 2021-08-10
> **Response to Review Comments**
>
> Thank you for your helpful insights and suggestions. Below, we discuss the main points you raised.
>
>
> * **IMPALA clarification.**
> We did not change IMPALA to learn a Q-function. The policy IMPALA learns is a softmax over logits $f(s,a)$. Actions with a higher logits value have higher chances of being selected, and since the goal is to learn a policy that leads to high cumulative rewards, logits can be interpreted as a Q-function. We thought that this notation would have been clearer and more familiar to an RL audience. To avoid confusion, we will change $Q(s,a)$ to $f(s,a)$.
>
>
> * **Formal definition of the whole problem.**
> Here is a formal definition of the problem.
> First, the agent explores a set of environments $E_1, E_2, \ldots, E_n$ governed by MDPs $\langle S_n, A, P, r_i, \gamma_i \rangle$, i.e., each environment has its own states, but the action space is the same in all of them, and all environments obey the same dynamics $P$ and the same intrinsic reward function $r_i$.
> At this pretraining time, the goal is to learn an exploration policy $\pi_{exp}$ that maximizes the sum of $\gamma$-discounted intrinsic rewards over all environments simultaneously.
> Then at transfer time, we build a policy $\pi_{task}$ based on $\pi_{exp}$.
> The agent is now placed in environments each governed by the classic MDP $\langle S,A,P,R,\gamma\rangle$ and learns $\pi_{task}$ to maximize extrinsic rewards over all environments independently.
> In our experiments, we use IMPALA to learn both $\pi_{exp}$ and $\pi_{task}$. IMPALA learns policies of the form $\pi(s,a) = \sigma(f(s,a))$, where $\sigma$ is the softmax function. The policy is trained to maximize a function representing the value of states $V(s)$, trained on the given rewards. In our framework, we combine the two policies as follows.
>     * During pre-training, by using intrinsic rewards we learn $V_i(s)$ and $\pi_{exp}(s,a) = \sigma(f_i(s,a))$.
>     * At transfer, by using extrinsic rewards we learn $V_e(s)$. The policy is $\pi_{task}(s,a) = \sigma(f_e(s,a) + f_i(s,a))$. The interestingness $f_i$ is transferred but not trained, i.e., it acts as fixed bias to encourage interaction.
> We will add this formal description to the paper.
>
>
> * **Do you keep the intrinsic reward after transfer?**
> No. At transfer, we keep only the exploration policy (logits function) and give the agent only extrinsic rewards. The reason is that a successful exploration policy (learned in our task-agnostic manner) should not need any further help from intrinsic rewards at transfer time. Adding intrinsic rewards at transfer time would be redundant. Furthermore, this way all algorithms have access to the same data at transfer, and what makes the difference is only the pre-trained policy. Therefore, we can accurately evaluate pre-trained policies.
> Also notice that we do not transfer the value function.
> We could think of transferring $V_i(s)$ as fixed bias as well, i.e., by having $V_e = V_{train} + V_i$. The policy would be trained on $V_e$ --the states value w.r.t. the given task-- where $V_i$ is fixed and only $V_{train}$ is updated.
> However, we believe it is more beneficial to isolate the exploratory component within the policy in order to keep the task-specific value function targeted on extrinsic rewards.
> By not transferring $V_i$, $V_e$ can be accurately trained on extrinsic rewards --that the agent will see often thanks to the pre-trained exploration policy. $V_e$, in turn, can make the policy greedy w.r.t. extrinsic rewards as $V_e$ is learned.
>
>
> * **Related to Figure 7: average task-specific reward (or solve rate) for CBET and each of the baselines in MiniGrid, averaged across all transfer tasks, to have a number to compare between the methods.**
> Thank you for this suggestion. We already have ‘solve rate’ (we call it ‘success’) at the beginning of transfer from *MultiEnv* in Figure 16 (supplementary). We will add:
>     * The same plot for transfer from *DoorKey* and *KeyCorridor*, and
>     * The same plot but at the end of transfer learning. Additionally, we will report the average success rate in the legend of the plot, for a quick comparison.
>
>
> * **It would be great to see the values of the state-count and change-count plotted for different environments and different settings.**
> We can add the trend of state and change counts (without resets) during pre-training, similarly to what we did for interactions and intrinsic rewards in Figure 6.
>
>
> * **Ablation study.**
> Please see our answer to R1’s comment (id: 3Bvy).
>
>
> * **Evaluation on another task with extrinsic after transfer. For Habitat, we showed only transfer at the end of pre-training but not the actual learning of an exploration policy on a navigation task.**
> Thank you for this suggestion. We can include a goal position in Habitat and give extrinsic rewards based on that. In the final version of the paper, we will evaluate full transfer for Habitat as we did for MiniGrid.
>
>
> * **BeBold and AGAC comparison.**
> Thank you for this suggestion. We can already compare C-BET against BeBold and AGAC by looking at some common environments they have been evaluated on and see after how many steps the task was solved).
>     * BeBold (Figure 3 of its paper): MultiRoom-N6 (10M), MultiRoom-N12-S10 (10M), KeyCorridorS3R3 (0.5M), ObstructedMaze-2Dlh (20M), ObstructedMaze-2Dlhb (40M).
>     * AGAC (Figure 2 of its paper): ObstructedMaze-2Dlhb (not solved after 100M).
>     * C-BET (Figure 7): KeyCorridorS3R3 (close to 0, almost solved already after transfer from MultiEnv), MultiRoom-N6 (10M from MultiEnv), MultiRoom-N12-S10 (10M from MultiEnv), ObstructedMaze-2Dlh (20M from MultiEnv), ObstructedMaze-2Dlhb (50M from MultiEnv).
> AGAC seems to perform significantly worse than both BeBold and C-BET.
> BeBold and C-BET perform very similarly, with C-BET achieving better results on KeyCorridor and BeBold on ObstructedMaze-2Dlhb.
>
>
> * **Figure 7, C-BET (Ours) vs Our Reward.**
> This was to stress that the C-BET key component is to transfer exploration, not only the introduction of the change-count. In the tabula-rasa setting, there is no transfer. Thus, with `Our Reward` we denote that we are only using only the change-count component of our approach, but not the transfer.
>
>
> * **Figure 7, RIDE cannot learn MultiRoom but it can in the original paper, why?**
> Please see our answer to R2’s comment (id baKZ).
>
>
> * **Smoothing: are evaluations very high variance? What do less smoothed curves look like?**
> Evaluations are performed online using the same samples collected for learning. We do not run separate episodes for the evaluations. Updates are performed when 3200 samples are collected (32 batches of 100-step rollouts). If among these samples there are states corresponding to the end of episodes, we use them to compute the return. Sometimes, especially in an environment with a long horizon (e.g., BlockedUnblockPickup or ObstructedMazes with 576 max steps), there are no ending samples in the batch, and the return cannot be computed. Therefore, we have --at best-- one data point every 3200 steps.
> This makes the curves a bit shaky, but the size of the shaded area (variance) is not different from the ones of smoothed plots.
>
>
> * **Line-specific comments.**
> We thank R4 for the detailed line-specific comments; we’ll be sure to incorporate everything in the final paper.

---

> > ### Comment · Reviewer_gCxP · 2021-08-19
> > **Response to rebuttal**
> >
> > Thanks for you detailed clarifications to my questions and suggestions. Given your comments in this response and your response to other authors, I believe my confusions have been clarified, and I believe the paper will be much improved with the improvements you've described here and elsewhere. Hence, I'm raising my score to an 8 - I believe the paper is an interesting, novel and relevant problem setting and presents a novel approach to solving it, while also bein rigorous in the evaluation of their and previous works to ensure that their contribution is robustly useful.

---

### Official Review · Reviewer_QKaX · 2021-07-09

**Rating:** 8
**Confidence:** 4

**Summary:**

This paper proposes a new paradigm for exploration based on transfer. They argue that we should not focus on exploration in isolated environments (it is not realistic for humans either), but instead focus on a sequence of tasks. More interestingly, they propose that we need to disentangle the exploration signal, into one agent-centric one based on visitation counts, but also a environment-centric one that identifies states where we get a change in the environment, and are thereby inherently interesting over tasks. They propose a new intrinsic reward that combines both properties, and test these in gridworld experiments and first-person navigation tasks, where they perform on par with or better than baseline exploration methods.

**Limitations And Societal Impact:**

Adequately addressed.

**Main Review:**

Strong:
* I really like the main idea: the fact that certain objects are interesting by definition, not in a single task, but over repeated tasks. This is indeed a novel perspective that I have not seen before, but makes a lot of sense.
* Extensive experimentation, interesting observations. For example that transfer using model characteristics is not efficient at all for transfer.

Weaker:
* I would like a more detailed comparison to Raileanu and Rocktäschel [36]. Since the claims in the introduction are quite strong (a completely new paradigm for exploration), you should motivate this carefully compared to related work (I think the statement is a bit strong).
* I am a bit in doubt about the “count resets”. I understand the reasoning, but it may actually be harmfull within a task (if there is one state we have not visited yet, then keeping the counts this may be rewarded highly, while with a reset we have to start from scratch again). I think the paper at least needs to discuss this, and a more elegant solution (which allows optimal exploration within the task as well as for transfer) would be preferable.
* I am a bit confused between c(s, a, s) (e.g. L215) which is a vector, and N( c) in Eq. 1. Is N( c) the sum over the vector?
* I am also confused about the panoramic images (L283-288). How do you align these for the equation in L 290 (otherwise the element-wise difference does not make sense right, if you have rotated your view every element will change).
* Fig 10: I find the Habitat interpretation really qualitative, the main result being “that C-Bet is the only policy visiting the right-hand corner”. Sure, but RIDE is the only one to visit the top-middle corner?

Summary:
I really like the idea of this paper, that task-agnostic exploration should be separated into generic environment-centric properties (objects that are always interesting) and agent-specific properties depending on the interaction history (such as novelty). I think this is an important insight, especially that one can estimate the former property during training. There are some smaller issue with the paper as mentioned above, but none of them is really big. The experiments are sounds as well, and indeed show that the proposed method works. I recommend this paper for acceptance, since I believe it will definitely interest the broader RL exploration community.


**Time Spent Reviewing:**

1.5

---

> ### Author Response · Authors · 2021-08-10
> **Response to Review Comments**
>
> Thank you for your helpful insights and suggestions. Below, we discuss the main points you raised.
>
>
> * **Comparison vs RIDE: why is C-BET a "new paradigm for exploration"?**
> We agree that RIDE is the main 'competitor' of our algorithm, given that both use a similar formulation (change for C-BET, impact for RIDE). This is why throughout our paper we have compared against RIDE on some of their environments using their code and implementations (and we have outperformed RIDE on most environments and in most cases by a significant margin).
> Philosophically, the two papers are quite different. RIDE performs task-dependent exploration (the extrinsic reward is always given) in tabula-rasa (nothing is shared across environments). On the other hand, C-BET explores the setting of transferring task-agnostic exploration: it uses change to transfer/generalize from one scene to another. RIDE does not attempt transfer/generalization at all. This is why we argue that ours is a new paradigm for exploration.
>
>
> * **More discussion on count resets. May it actually be harmful for some tasks? Is there a more elegant solution?**
> Thank you for this suggestion. We agree that there are some shortcomings of episodic resets. First, episodic resets have a ‘fixed memory’: by starting an episode with zero-counts the agent ‘forgets’ everything that happened in the past, thinking that every state and change it sees is new. Second, many real-world problems do not have episodes, and episodic resets cannot be applied in infinite-horizon problems, where the agent/environment is never reset.
> Therefore, we also evaluate an alternative reset strategy: random count resets, i.e., counts are reset at any given time step with probability $p$. When a new episode starts, counts may not be reset yet so the agent remembers what it has visited before. Then, on average more common states and changes will have higher count more often, and the agent will correctly prioritize rarer ones. As for the reset probability, we suggest $p = 1 − \gamma_i$, where $\gamma_i$ is the intrinsic reward discount factor.
> This is a fitting choice because in an MDP the sum of discounted rewards can be interpreted as the expected sum of undiscounted rewards if every time step had a $1 − \gamma_i$ probability of terminating. Intuitively, this means that the discount factor $\gamma_i$ implies a ‘life expectancy’ of $1/(1 − \gamma_i)$ steps: the higher the discount factor, the longer the policy will explore.
> We note that C-BET still outperforms all baselines in all environments with random resets. Yet, this formulation with random resets is more general and can be applied to more environments. We will add a section to discuss this in the final version of the paper.
>
>
> * **Confusion about $c(s,a,s')$.**
> $c(s,a,s')$ is a vector used to uniquely identify a change. You can think of it as a Python dictionary key. Every time we encounter a new change (a vector), we look for it in the dictionary and increase it, or set it to one if it is a new change. The dictionary, thus, counts occurrences of keys (changes). We do the same for state counts.
>
>
> * **Confusion about panoramic images.**
> The first image is always with respect to the North. Then we move the camera clockwise and concatenate the images. The resulting (64,64,3,4) panoramic images are what we count (using pseudocounts).
>
>
> * **Figure 10, “C-BET is the only policy visiting the right-hand corner, but RIDE is the only one to visit the top-middle corner”.**
> In the offline testing performed in Habitat, the agent always starts at the same position (top-left) and explores the environment for a fixed amount of steps (500 for big rooms, 200 for small rooms). The best performing exploration agent, thus, is the one that visits more states within the time limit. And C-BET quantitatively visits more states than RIDE and other baselines (as demonstrated by Figure 8). It is also harder for the agent to go all the way to the right corner rather than the middle section, which is closer to the starting point. To visit the top-right corner, the agent must navigate almost the entire room.

---

### Official Review · Reviewer_baKZ · 2021-07-15

**Rating:** 8
**Confidence:** 4

**Summary:**

This paper focuses on task-agnostic exploration in RL, with two key ideas:
- introducing a new evaluation method for task-agnostic exploration, by measuring "how well" the learnt exploration policy behaves in new tasks; "how well" can either be measured based on extrinsic return in the withheld task, or on exploration-related quantities like coverage
- proposing a new count-based intrinsic reward that is shown to outperform other baselines in the transfer task in procedurally-generated gridworlds, and in the coverage task in 3D scenes from Habitat.

**Limitations And Societal Impact:**

There is a clear and fair acknowledgement of limitations in an explicit section, which I really appreciated.

**Main Review:**

The article is well written with a wealth of experimental results, as well as thorough in the Related work section.

Originality:
- The idea of measuring how well a policy transfers is not new, but I have not seen it applied specifically to evaluate exploration policies.
- As far as I know, combining state visitation count (already established in the literature) with change count has never been published.

Contribution/significance: I think good enough, with one "medium" and one minor contribution.
- The first problem addressed by the paper -- evaluating task-agnostic exploration -- is an important one in the community, and I think the proposed protocol has merits. There are a few limitations that I think currently limit the applicability of the method, but they seem somewhat easy to overcome and are clearly identified in the paper. To me this is a significant enough contribution.
- The proposed C-BET exploration algorithm seems to achieve good performance according to that metric. I have to say I'm always a bit wary of papers that both introduce a new benchmark and a new algorithm that outperforms its competitors on that benchmark, but in this instance the relatively thorough set of baselines offsets this issue. However I'm not sure how generally applicable C-BET is (both in terms of scalability and applicability).

Questions:
- How scalable is C-BET when the state-action space grows? In the Habitat experiments, the authors had to use a very coarse action space with 3 actions (thus drastically reducing the complexity from a 3D env to a gridworld with rich observations). Does C-BET still perform reasonably well with, eg, 30° vertical and horizontal rotations?
- How does C-BET perform in environments with intra-episode stochasticity (eg, noisy tv)? The 1/sqrt(N(c)) will ultimately compensate for this if the episode is long enough, but because counts are reset every episode, the additional |A| factor (|S||A| state-action pairs to visit instead of just |S| states) seems like it can significantly hurt performance in practical setups.
- Figure 7: Shouldn't the results on MultiRoom-N12-S10 ("no transfer" row) match those of Fig 3h in https://arxiv.org/pdf/2002.12292.pdf? RIDE should get to ~0.6 in 25M frames.
- Figure 9: it's not clear to me how to interpreter error bars. Is the colored bar showing the average and the top of the error bar the maximum over 5 seeds? If so, what about the minimum?
- Section 5, "advantage" paragraph (and scattered throughout the rest of the paper): I'm not sure I see how C-BET accounts for "rarity" of change, since eg the agent's position is presumably part of the state, and C-BET doesn't have any form of (action, state) embedding that would be able to learn translational (or other) invariances? Presented with an environment with, eg, N on/off switches that are part of the state but do nothing interesting, won't C-BET be tempted to explore all 2^N combinations?

Extra comment: unless negative rewards exist in MiniGrid, something seems off with the smoothing of curves (eg Fig7)

**Time Spent Reviewing:**

5

---

> ### Author Response · Authors · 2021-08-10
> **Response to Review Comments**
>
> Thank you for your helpful insights and suggestions. Below, we discuss the main points you raised.
>
>
> * **Scalability. In particular, what if we use different action discretization in Habitat?**
> C-BET scalability is tied to the state space only. Increasing the size of image observations or using continuous state vectors would affect C-BET performance (depending on how fine-grained pseudocounts are). On the contrary, increasing the number of actions or size of the action step (30 vs 90 degrees) affects the scalability of all approaches similarly, including both baselines and C-BET. C-BET counts states and state changes and thus is not affected specifically by the action space, at least not more than any other method.
>
>
> * **Evaluation with stochasticity.**
> Thank you for this suggestion. Very recently, the authors of RIDE have released the code for their noisy environment (*MultiRoomNoisyTV*), and we will evaluate all algorithms on it in the supplementary material.
>
>
> * **Figure 7, RIDE cannot learn MultiRoom but it can in the original paper, why?**
> First, we want to highlight that we have used both the MiniGrid and RIDE baseline directly from their released code (no changes to code).
> We believe the key reason for the difference is RIDE’s sensitivity to hyperparameters. RIDE needed differently tuned coefficients even for different versions of the MultiRoom. In the original paper, they used tuned intrinsic reward coefficients for each environment. Other baselines (RND, Count, Curiosity) all used the same coefficient, but RIDE performed differently depending on the coefficient.
> In our paper, we used an intrinsic reward coefficient of 0.1 for RIDE. In its original paper, this coefficient is used for MultiRoomN7S4, MultiRoomN10S4, KeyCorridorS3R3. For MultiRoomN7S8, MultiRoomN10S10, MultiRoomN12S10, and ObstructedMaze2Dlh it used 0.5.
> Finally, we highlight that C-BET worked well in all environments with the same coefficient
>
>
> * **Figure 9, error bars.**
> Error bars show standard deviation. The colored bar shows the mean and the top of the error bar represents (mean + std).
>
>
> * **How does C-BET account for changes in the environment if the agent is part of it?**
> In our experiments, the agent’s position is not part of the observation, since observations are partial observations of the environment (both in MiniGrid and Habitat). Regardless, C-BET emphasizes environment changes even when the agent is part of the observation. Let us clarify it with an example.
> Consider the *DoorKey* environment, with 1 key and 1 door. Say that at first the agent moves everywhere and visits all cells. Every state was also a different change, as you said. Then it picks up the key and keeps moving again. States are still new (because the agent is carrying the key, which is part of the state), but changes are not new (moving through the grid with/without the key is the same in terms of environment changes). Counts related to move/turn will thus quickly get high counts, and the agent will be more rewarded for interacting with the key (or --even better-- with the door). And this is exactly the behavior we are looking for.
> We agree that using an (action, state) embedding rather than raw observations could be better but even with full raw observations C-BET still favors environmental changes.
> Finally, let’s consider your example of $N$ switches. C-BET values rare changes and rare states altogether. Considering only states would lead to the behavior you described: the agent will try all $2^N$ combinations. Change counts, though, care only about the $N$ possible changes that can happen. If there are 3 switches, the change `000 -> 001` is identical to `010 -> 011`, even though states are different. The introduction of change counts, thus, mitigates what would happen otherwise considering only state counts. Thanks to change counts, in fact, the agent will prioritize flipping all $N$ switches first, rather than trying all $2^N$ combinations.
>
>
> * **Unless negative rewards exist in MiniGrid, something seems off with the smoothing of curves (e.g., Figure 7).**
> The variance shading is symmetric, so even if the shading goes below 0, that does not imply that any seed had negative rewards. It simply means that there was more variance in the upper parts of the curve.

---

### Official Review · Reviewer_3Bvy · 2021-07-28

**Rating:** 9
**Confidence:** 4

**Summary:**

* This paper challenges the contemporary experimental setup of exploration algorithms, and suggests that we should not start exploring in new environments tabula-rasa, and instead use knowledge derived from prior experience to bootstrap exploration.
* The paper also identifies issues with using existing tabula-rasa exploration algorithms for the new setting, and propose a new method C-BET to overcome those shortcomings.
* C-BET and a bunch of existing exploration algorithms are benchmarked on newly constructed transfer exploration setups in the MiniGrid and Habitat environments.

**Limitations And Societal Impact:**

Yes

**Main Review:**

This paper offers a fresh perspective on training and evaluation exploration algorithms for RL environments. The current paradigm in training exploration methods is to start "tabula-rasa" in each environment, with no pre-conceived knowledge about the world. This means that the agent has to blindly try-out and explore a combinatorically large state-action space which then leads to requiring billions of frames or years of experience to solve sparse reward problems, something that's impractical for real-world usage. Not to mention the inherent safety issues involved with tabula-rasa exploration. The paper proposes a new experimental setup to evaluate exploration algorithms: the algorithms are first pre-trained in a task-agnostic manner on some environments and then transferred to unseen environments.

The writing in the paper is clear throughout and it does good job of explaining why this different experimental setup makes sense. I also appreciate the generous use of figures to make the exposition clear. One of the most important contributions of this paper is the construction of new experimental setups for exploration transfer, and benchmarking existing methods on them. These experimental setups cover MiniGrid (a suite of GridWorld environments) and Habitat (a suite of visual 3D environments), thus covering a wide spectrum of scenarios.

Its interesting to see curiosity like methods fail during transfer, but somewhat obvious in hindsight. After learning a predictive model of the world, similar states are not so surprising anymore even though they might still be inherently interesting in the new environment. C-BET tries to fix this by using a pseudo count of interesting changes along with count resets every episode. The results on pre-training and task-transfer are quite compelling. The analysis in the supplementary material is quite informative and its interesting to see that a simple trick like count resets has such a large impact.

Overall, I really like this paper, the ideas and perspective on exploration are original, presented well and are of great significance. I would have recommended acceptance even without the introduction of a new method, but that is an added bonus. I do have some comments / questions though and it would great if the authors can expand on it:
* Can you benchmark / ablate the impact of *change* pseudo counts? Specifically, running your "Count" baseline along with episode resets, and comparing with C-BET (count + change + episode resets) would clarify this.
* In the Habitat experiments, the use of panaromic images seems like leveraging access to privileged informtation that other methods don't have. Have you tried experimenting with using egocentric images for pseduo-counts and the change encoding? Does it lead to a significant performance drop?

**Time Spent Reviewing:**

5

---

> ### Author Response · Authors · 2021-08-10
> **Response to Review Comments**
>
> Thank you for your helpful insights and suggestions. Below, we discuss the main points you raised.
>
> * **Ablation study**.
> `C-BET` introduces count resets and a novel count (change-count). Study these components separately. State-count-only + resets would correspond to `Count` with resets.
> We agree and have performed your suggested comparison. The results show that count resets significantly increase the performance of `Count`, yet `C-BET` still performs best. In particular, the success rate evaluation after pre-train on *MultiEnv* (Figure 16) shows that `C-BET` achieves ~49% success rate (averaged over all 10 test environments), while `Count with resets` ~29%. Other baselines achieve only 1-2%.
> This is not surprising, considering the problem of vanishing reward discussed in the paper, which is the main reason why other baselines fail. We will include these results in the final version of the paper, as well as the full learning after transfer evaluation.
>
> * **Investigate Habitat with frames rather than panoramic images for change-counts.**
> We are happy to investigate Habitat with frames rather than panoramic images. We will add this experiment in a camera-ready version. That said, we did do a similar study for MiniGrid. In MiniGrid, both panoramic images and the frames worked well (panoramic worked slightly better).

---

> > ### Comment · Reviewer_3Bvy · 2021-08-23
> > **Response to rebuttal**
> >
> > Thanks a lot for adding the count+reset baseline, it adds more clarity on the role of resets. In light of the ablation, I would encourage you to highlight the importance of resets even more in the text.

---

### Author Response · Authors · 2021-08-10
**To All Reviewers and ACs**

We thank the reviewers for their feedback and helpful suggestions. We are pleased to see that all reviewers appreciate the core idea of our paper --a novel perspective on training/transfer exploration policies for RL, with a focus on how to explore in a task-agnostic manner-- and the thorough evaluation against different baselines on many environments. We are also happy to see that reviewers believe our idea of ‘environment-centric’ exploration is intuitive and novel.

We have replied to each reviewer separately to address some specific clarifications. In this comment, we recap the main suggestions/questions presented by the reviewers and discuss what we will introduce in the final version of the paper.

* **Count resets**
R3 (QKaX) asked for a more in-depth discussion of episodic resets and suggested that there may be scenarios in which these resets could be harmful. We discuss some of these scenarios below and also investigate an alternative reset strategy: random resets. In random reset, counts are reset at every step with probability $1 − \gamma$. One key advantage of random reset is that it can also be applied in continual learning setup when the concept of ‘episode’ might not be there. For more details, please refer to our reply to R3 (QKaX).
C-BET w/ random resets outperforms all baselines and transfers to all environments. We will also add a section to discuss random resets in the final version of the paper.


* **Ablation study**
R1 (3Bvy) and R4 (gCxP) asked for an evaluation of C-BET with only change-counts and with only count-resets. We have done that, and C-BET with both components performs the best. The biggest improvement is given by the count resets, but without including the change-count we cannot achieve the same performance. We will include these ablations in the final version of the paper.


* **A more formal description of C-BET framework**
Following R4's (gCxP) request, we have added formal details about the pre-training and transfer setup, explicitly discussing what we learn (policies and value functions) and what we transfer. For more details, please refer to our reply to R4 (gCxP).


* **Additional evaluations**
We will add some additional experiments to the camera-ready version of the paper, as suggested by the reviewers (please note that some of these experiments use 30 train/test cases with 50M+ steps and therefore have not finished).

---

### Decision · Program_Chairs · 2021-09-27

**Decision:**

Accept (Oral)

**Comment:**

This paper introduces a new problem setup for exploration in RL, which is to allow the agent to learn an exploratory policy in the absence of reward/task and evaluate how quickly the agent learns when the task is given in the transfer phase. All of the reviewers highly appreciated this problem setup and agreed that this setup is reasonable and deserves more attention, though a similar setup has been considered in the past [1], which should be cited and discussed in the paper. In addition, the paper proposes a new count-based exploration method that encourages rare state changes and proposes to reset count at every episode to overcome the vanishing intrinsic reward issue specifically for the new proposed problem setup. The results on MiniGrid and Habitats environments show that the proposed method performs better than baselines in both the exploration phase and the transfer phase. The authors promised to address several minor questions around count reset and ablation in the revision. Overall, the new problem setup and the proposed method would be quite interesting to the research community. Thus, I recommend acceptance for this paper.

[1] Rajendran et al., How Should an Agent Practice?